# Dominant *Vibrio cholerae* phage exhibits lysis inhibition sensitive to disruption by a defensive phage satellite

**Stephanie G Hays[1], Kimberley D Seed[1,2]\***

[1]Department of Plant and Microbial Biology, University of California, Berkeley, United States; [2]Chan Zuckerberg Biohub, San Francisco, United States

**Abstract** Bacteria, bacteriophages that prey upon them, and mobile genetic elements (MGEs) compete in dynamic environments, evolving strategies to sense the milieu. The first discovered environmental sensing by phages, lysis inhibition, has only been characterized and studied in the limited context of T-even coliphages. Here, we discover lysis inhibition in the etiological agent of the diarrheal disease cholera, *Vibrio cholerae,* infected by ICP1, a phage ubiquitous in clinical samples. This work identifies the ICP1-encoded holin, *teaA,* and antiholin, *arrA,* that mediate lysis inhibition. Further, we show that an MGE, the defensive phage satellite PLE, collapses lysis inhibition. Through lysis inhibition disruption a conserved PLE protein, LidI, is sufficient to limit the phage produced from infection, bottlenecking ICP1. These studies link a novel incarnation of the classic lysis inhibition phenomenon with conserved defensive function of a phage satellite in a disease context, highlighting the importance of lysis timing during infection and parasitization.

## Introduction

Following the discovery of bacteriophages (*D'Herelle, 1917*; *Twort, 1915*), *Escherichia coli*'s T1 through T7 phages were widely accepted as model systems (*Keen, 2015*) and T-even phages were used to determine mutation manifestation at the molecular level, gene topology, and that fact that nucleic acids are decoded in triplets (*Benzer, 1961*; *Crick et al., 1961*). Early geneticists observed an interesting phenotype in rapid lysing T-even mutants, called *r* mutants, which produce plaques with clear edges while plaques of wild type (WT; all acronyms are expanded in *Table 1*) T-even phages have fuzzy edges (*Hershey, 1946*; *Paddison et al., 1998*). Edge fuzziness is the consequence of inhibited cell lysis triggered by the adsorption of additional phage after initial infection. This 'superinfection' also stabilizes infected cells as measured by optical density (*Doermann, 1948*). The phenomenon, termed lysis inhibition (LIN), is significant for two reasons (*Figure 1A and B*): it allows for prolonged production of progeny phage resulting in larger phage bursts, and it protects progeny phage from adsorbing to infected cells, which are not productive hosts for secondarily adsorbed phages (*Abedon, 1990*; *Abedon, 2019*; *Doermann, 1948*). Consequently, LIN is considered an important adaptation in environments where host bacteria are scarce but free virions are plentiful (*Abedon, 1990*; *Abedon, 2019*). Despite being discovered over half a century ago, LIN has only been well characterized in T-even coliphages where it is mediated by holins and antiholins (*Chen and Young, 2016*; *Paddison et al., 1998*; *Ramanculov and Young, 2001*). Holins are the first step in canonical holin-endolysin-spanin lysis systems in Gram-negative bacteria (*Cahill and Young, 2019*; *Young, 2014*). Holins accumulate in the inner membrane until they trigger, making holes that enable endolysin digestion of the peptidoglycan after which spanins fuse the inner and outer membranes to complete cell lysis. During lysis inhibition, antiholins inhibit holin triggering thereby stopping the progression towards lysis (*Chen and Young, 2016*; *Paddison et al., 1998*; *Ramanculov and Young, 2001*).

**\*For correspondence:**
kseed@berkeley.edu

**eLife digest** Bacteriophages, or phages for short, are viruses that infect bacteria, take over the molecular machinery inside the bacterial cells and use it to make more copies of themselves. The bacteriophages then break open, or "lyse", the bacterial cell, releasing the viral copies into the environment, ready to infect more bacteria nearby.

Hays and Seed set out to understand how the timing of lysis can impact the bacteriophage, using the bacterium *Vibrio cholerae* – which causes cholera – and its bacteriophage called ICP1. This analysis revealed that the ICP1 phage uses a gene called *teaA* as the first step in the lysis of bacterial cells. The ICP1 phage can also delay that lysis with a second gene called *arrA*. This "lysis inhibition" gives the bacteriophages more time to make copies of themselves inside the bacterium, so even more are released when the cell finally breaks open.

Hays and Seed also found that the *Vibrio cholerae* cells can defend themselves against lysis inhibition using a single gene called *lidI*. This gene is part of a system that defends against bacteriophage attack called the PLE, which consists of several genes of previously unknown function. Hays and Seed saw that the *lidI* gene disrupts lysis inhibition, speeding up the bursting of infected bacterial cells, which in turn decreases the number of bacteriophages produced from each infected cell.

Lysis inhibition had previously only been observed in the bacterium *Escherichia coli*. Now that researchers know that ICP1 bacteriophages also delay lysis in *Vibrio cholerae*, this might lead to more studies exploring this process in samples from cholera patients. Further studies could test to see if the phenomenon of lysis inhibition may also exist in yet more bacterial species.

Phages are abundant in natural environments including marine ecosystems (*Middelboe and Brussaard, 2017*) and gut microbiomes (*Shkoporov and Hill, 2019*), potentially making LIN a pertinent state for phages infecting many different bacteria, including *Vibrio cholerae*. *V. cholerae* poses a substantial global health burden as the causative agent of the diarrheal disease cholera (*Ali et al., 2015*). In both aquatic reservoirs and stool samples from cholera patients, *V. cholerae* co-occurs with

**Table 1.** Acronyms.
All the acronyms used in this work are listed in alphabetical order.

| Acronym | Meaning |
| --- | --- |
| DiOC$_2$(3) | 3,3'-diethloxacarbocyanine iodide |
| DNP | 2,4-dinitrophenol |
| EOP | efficiency of plaquing |
| EV | empty vector |
| gp | gene product |
| IM | inner membrane |
| LIN | lysis inhibition |
| MGE | mobile genetic element |
| MOI | multiplicity of infection |
| MOSI | multiplicity of superinfection |
| OD | optical density |
| OM | outer membrane |
| ORF | open reading frame |
| PG | peptidoglycan |
| PLE | phage-inducible chromosomal island-like element |
| SaPI | <u>S</u>taphylococcus <u>au</u>reus <u>p</u>athogenicity <u>i</u>sland |
| WT | wild type |

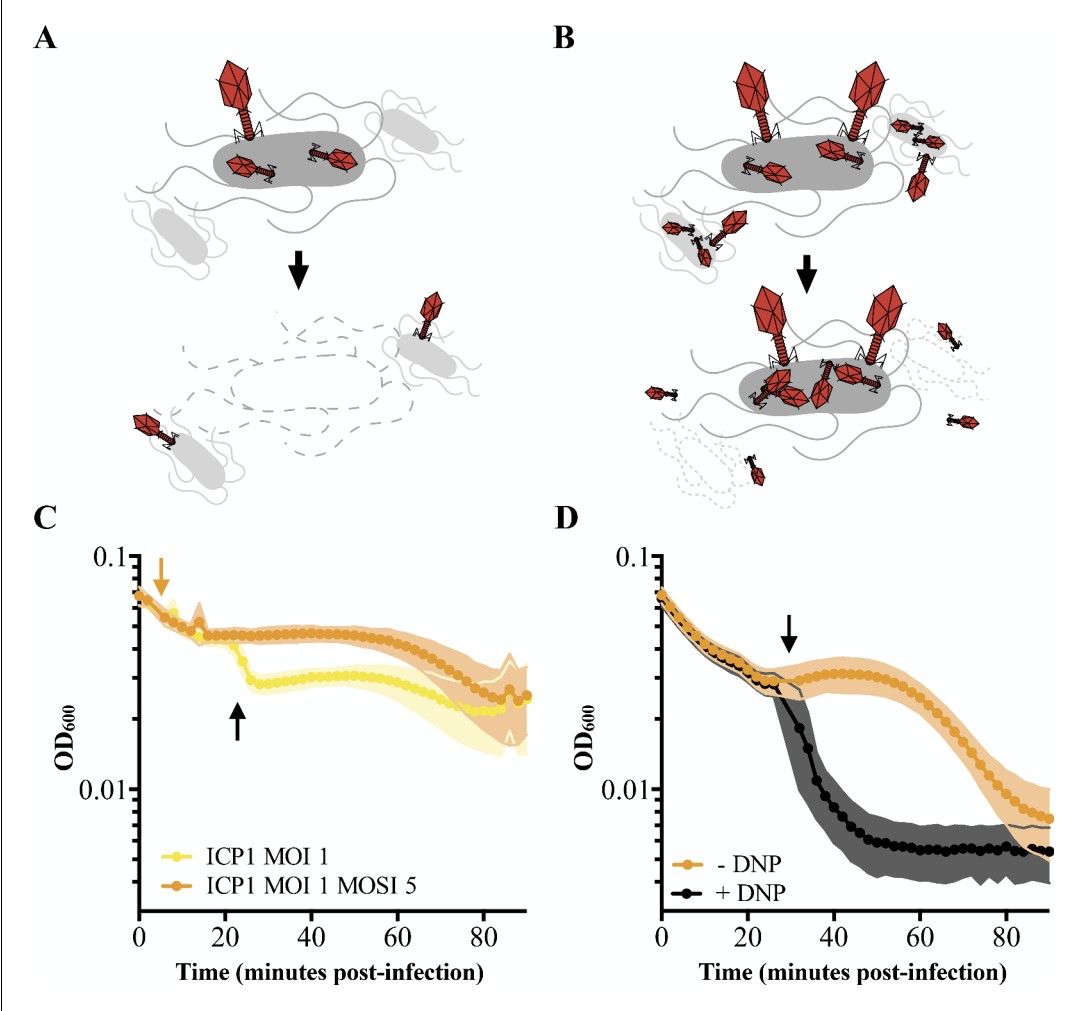

**Figure 1.** Characterizing lysis inhibition. Schematic of T4 infection of *E. coli*. (**A**) At low multiplicities of infection (MOI), available *E. coli* are readily infected by T4 (red). Under these conditions, an infected cell is hijacked to produce progeny phage and lyses releasing the phages into the environment where they go on to infect neighboring cells. (**B**) At high MOIs, phage outnumber the hosts resulting in initial infection by phage followed by secondary superinfection (second phage infecting the same cell). Superinfection delays lysis through LIN, enabling the production of more virions and protecting progeny phage from an environment devoid of uninfected hosts. (**C**) ICP1 exhibits LIN at intermediate multiplicities of infection and upon superinfection. PLE (-) *V. cholerae* infected with ICP1 at MOI = 1 demonstrate a lysis event 20 minutes post-infection (black arrow) before the optical density (OD$_{600}$) stabilizes. Superinfection (orange arrow) of a culture four minutes post-infection with ICP1 at MOI = 1 with ICP1 at a multiplicity of superinfection (MOSI) of 5 triggers lysis inhibition and stabilizes the OD$_{600}$ before a lysis event can occur. Data from three biological replicates are shown. (**D**) ICP1 LIN is sensitive to chemical collapse. PLE (-) *V. cholerae* infected with ICP1 at MOI = 5 maintain OD$_{600}$ for an extended period but the OD$_{600}$ collapses when 2,4-dinitrophenol (DNP) is added (black arrow). Data from four biological replicates are shown. For all graphs, points show the average of replicates; shading shows the standard deviation.

The online version of this article includes the following source data for figure 1:

**Source data 1.** This spreadsheet contains the data used to create *Figure 1C and D*.

predatory phages. Several studies have implicated *V. cholerae* phages in playing a role in modulating cholera outbreaks (*D'Herelle and Malone, 1927*; *Faruque et al., 2005a*; *Faruque et al., 2005b*; *Jensen et al., 2006*) leading to proposals of phage cocktails as prophylactics to curb cholera transmission (*Yen et al., 2017*). The predominant phage in cholera patient samples is ICP1, a lytic myovirus (*Seed et al., 2011*) that is locked in a dynamic arms race with no clear winner as both *V. cholerae* and ICP1 continue to be isolated from patients in the cholera endemic region of Bangladesh (*Angermeyer et al., 2018*; *McKitterick et al., 2019a*; *McKitterick et al., 2019a*; *Seed et al., 2011*). Added into the evolutionary fray is a parasitic phage satellite called PLE (phage-inducible chromosomal island-like element) found integrated in the chromosome of clinical *V. cholerae* isolates

(*McKitterick et al., 2019b*; *O'Hara et al., 2017*; *Seed et al., 2013*). Previous analysis of isolates dating back to 1949 revealed a succession of five distinct PLEs (PLE 1 through PLE 5 of which PLE 1 is the most recently circulating PLE) in *V. cholerae* which possess shared genomic architecture. Each PLE provides *V. cholerae* with a clear fitness benefit in the defense against ICP1 as PLE abolishes phage production (*O'Hara et al., 2017*). Upon infection, PLE excises from the bacterial chromosome, harnessing an ICP1-encoded protein as the trigger (*McKitterick and Seed, 2018*), replicates using both PLE and ICP1-encoded products (*Barth et al., 2020*; *McKitterick et al., 2019a*), and accelerates cell lysis after forming particles hypothesized to be made by hijacking ICP1 structural components to transduce the PLE genome to naïve recipient cells (*O'Hara et al., 2017*). As no infectious ICP1 are produced, PLE defends populations of *V. cholerae* from ICP1 attack, functioning as an abortive infection system. In the face of this anti-phage element, ICP1 acquired a Type I-F CRISPR-Cas system to target PLE in a sequence-specific manner, restoring ICP1 progeny phage production and overcoming PLE (*McKitterick et al., 2019b*; *Seed et al., 2013*). While the full extent of ICP1, PLE, and *V. cholerae* interactions are unknown and continuing to evolve, efforts to understand how PLEs restrict ICP1 have yet to identify any single PLE open reading frame necessary for inhibition (via testing of PLE-encoded *repA*, which is necessary for PLE replication [*Barth et al., 2020*] and *int*, which is required for PLE excision [*McKitterick and Seed, 2018*]). From the only characterized processes, namely excision and replication, it is clear that PLE requires phage-encoded products, consequently depending on ICP1 for horizontal transmission; however, uncharacteristic of other phage satellites like the well characterized Staphylococcus aureus pathogenicity islands (SaPIs) which decrease but do not eliminate the production of progeny virions, PLE completely abolishes ICP1 production – a balancing act that likely requires the exploitation of select products at exact times during the 20 minutes before PLE-mediated accelerated cell lysis.

The accelerated cell lysis program in *V. cholerae* harboring PLE led us to investigate the prolonged infection of ICP1 in strains without PLE where we discovered that ICP1 exhibits lysis inhibition. In this work we report the first mechanistic characterization of archetypal LIN outside of *E. coli* and we reveal ICP1 LIN mechanisms in *V. cholerae* by identifying previously uncharacterized ICP1 genes with holin and antiholin activity, termed *teaA* and *arrA* respectively. Subsequently, we discovered a single PLE-encoded gene we call *lidI* for lysis inhibition disruption that is sufficient to collapse ICP1-mediated lysis inhibition. All PLEs encode LidI, highlighting a conserved strategy PLEs may use to antagonize an aspect of the phage lifecycle not previously known to be targeted by parasitic satellites. While we cannot be sure of LidI function in the context of the PLE, it alone is sufficient to decrease the yield of ICP1 from infected *V. cholerae* and impose an evolutionary bottleneck on phage populations.

## Results

### ICP1 exhibits lysis inhibition

After infection at low multiplicity of infection (MOI; MOI = 0.1), ICP1 completes virion production within 20–25 minutes in PLE (-) *V. cholerae* (*O'Hara et al., 2017*). This timeframe is markedly abbreviated with respect to the 90 minutes that pass before visible lysis of the same host infected at a high MOI (MOI = 5) (*O'Hara et al., 2017*). This incongruity prompted us to test ICP1 infections at intermediate multiplicities of infection. When infecting PLE (-) *V. cholerae* with ICP1 at MOI = 1, we observed an early lysis event 20 minutes post-infection after which the optical density of the culture stabilized (*Figure 1C*). Such lysis kinetics are consistent with canonical LIN by T-even coliphages wherein a portion of infected cells release progeny phages triggering LIN in the remaining population (*Doermann, 1948*). These similarities led us to hypothesize that ICP1 exhibits LIN in *V. cholerae*.

During canonical LIN, superinfection, the secondary adsorption of phages after initial phage infection, stabilizes the optical density of infected *E. coli* cultures because cells stay intact instead of lysing (*Figure 1B*). To determine whether superinfection by ICP1 of PLE (-) *V. cholerae* shares this characteristic, we infected cultures with ICP1 (MOI = 1), let phage adsorb for four minutes, then superinfected the culture with ICP1 (multiplicity of superinfection; MOSI = 5). As expected of a phage exhibiting LIN, the culture optical density was stabilized, eliminating the early lysis event (*Figure 1C*).

Previously characterized lysis inhibition in *E. coli* is sensitive to the membrane proton motive force and is disrupted by the addition of energy poisons. One such poison, the ionophore 2,4-dinitrophenol (DNP), collapses the proton motive force and subsequently disrupts LIN by T-even phages, causing rapid lysis of infected *E. coli* (*Abedon, 1992*; *Heagy, 1950*). To test if ICP1 LIN is similarly linked to proton motive force, we exposed PLE (-) *V. cholerae* infected at a high MOI (MOI = 5) to 2,4-dinitrophenol (*Figure 1D*) and observed the expected crash in optical density, further supporting the conclusion that ICP1 exhibits LIN in *V. cholerae*.

## ICP1 lysis inhibition is mediated by the putative holin, *teaA*, and the putative antiholin, *arrA*

Although genomes of ICP1 isolates from cholera patient stool are abundant, the genes involved in ICP1-mediated lysis have not been identified. To characterize the mechanism underlying ICP1 lysis inhibition, we endeavored to find ICP1's holin and antiholin (*Figure 2A*). Holins are diverse proteins, however, they all include at least one transmembrane domain (*Wang et al., 2000*). Further, we hypothesized that the holin would be conserved in isolates over time because lysis timing is key to

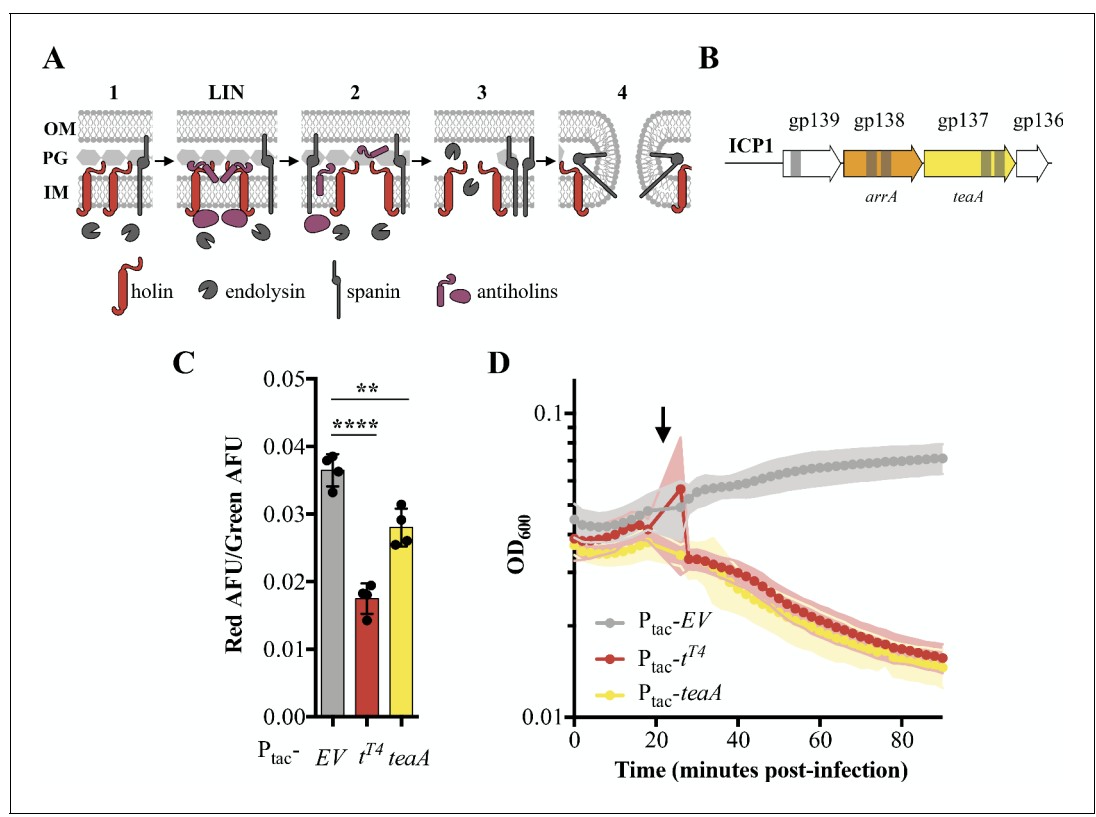

**Figure 2.** Identification and characterization of ICP1's holin TeaA. (A) Schematic of canonical T4 lysis in *E. coli*. Lysis occurs in four steps with a potential delay caused by lysis inhibition. Step one: Phage encoded proteins including holins, endolysins, and spanins accumulate in the cell. In the event of superinfection, antiholins halt the lysis program causing lysis inhibition (LIN). Step two: Holins are triggered collapsing the proton motive force and forming holes in the inner membrane (IM) releasing endolysins to the periplasm. Step three: The peptidoglycan (PG) is degraded by endolysins. Step four: Spanins fuse the inner and outer membrane (OM). (B) Transmembrane domains (grey bars) were predicted in three gene products (gp) in a conserved locus of ICP1. (C) Relative state of the proton motive force as measured by the ratio of red to green arbitrary fluorescence units (AFU) from $DiOC_2(3)$ after ectopic gene expression. Points represent individual replicates; bar height is the average; error bars display the standard deviation of samples. Significance was calculated via one-way ANOVA followed by Dunnett's test. ****$p \leq 0.0001$; **$p \leq 0.01$. (D) Chemical triggering with 2,4-dinitrophenol (arrow) after ectopic gene expression as measured by optical density ($OD_{600}$). Points show the average of four replicates; shading shows the standard deviation.

The online version of this article includes the following source data for figure 2:

**Source data 1.** Fluorescence Source Data.
**Source data 2.** DNP Source Data.

phage fitness. As a result we narrowed our search to gene products containing a predicted transmembrane domain that were conserved in previously analyzed ICP1 isolates (*Angermeyer et al., 2018*) leaving us with three candidate gene products: Gp137-Gp139 (*Figure 2B*; further described in *Supplementary file 1*). Analysis of these proteins using remote homology and synteny via Phagonaute identified the DUF3154 Pfam (reclassified as GTA_holin_3TM (PF11351); *Delattre et al., 2016*; *El-Gebali et al., 2019*; *Sonnhammer et al., 1997*) in ICP1 Gp137, which we have since named TeaA.

In *E. coli,* canonical holins including T4's T-holin, accumulate in the membrane until they collapse the proton motive force and form pores or are triggered by an energy poison. Premature holin triggering results in loss of viability and commits the cell to eventual lysis, decreasing the optical density (*Garrett and Young, 1982*; *Josslin, 1971*). To experimentally test TeaA for holin activity, we exogenously expressed *teaA* in PLE (-) *V. cholerae* in the absence of phage and probed its ability to collapse the proton motive force and be triggered by an energy poison. We measured proton motive force using 3,3'-diethloxacarbocyanine iodide ($DiOC_2$(3)), a fluorescent green membrane stain that forms red fluorescent aggregates in the presence of intact proton motive force (*Kirchhoff and Cypionka, 2017*; *Novo et al., 1999*). Upon induction, T-holin[T4] and TeaA decreased the red fluorescence, consistent with holin activity, while the empty vector (EV) did not (*Figure 2C*). Holin activity by TeaA was further demonstrated by the rapid decrease in optical density after 2,4-dinitrophenol addition which was comparable to the decrease in optical density observed with T-holin[T4] expressing *V. cholerae* after 2,4-dinitrophenol treatment (*Figure 2D*). Interestingly, though endolysin activity is described as necessary in holin-endolysin-spanin systems (*Young, 2014*), the holin T[T4] or TeaA alone is enough to lyse *V. cholerae* under laboratory conditions. The similarities between TeaA activity and T4's T-holin motivated the naming of *teaA* for the gene's T-holin-esque activity.

Next, we sought to identify an antiholin in ICP1. Antiholins can interact with holins within the inner membrane (*Moussa et al., 2012*; *Young, 2013*), periplasm (*Tran et al., 2005*), or cytoplasm (*Chen and Young, 2016*). Because antiholins interacting directly with holins within the membrane contain transmembrane domains and previously characterized periplasmic antiholins contain transmembrane domains for tethering to the membrane and subsequent release into the periplasm (*Tran et al., 2007*), we continued to focus on the transmembrane domain-containing proteins in ICP1. Given that TeaA is conserved in ICP1 isolates, and lysis timing, which is fine-tuned by antiholins, is critical, we expected antiholins would also be conserved. Antiholins are often found near holin genes, in many cases utilizing an alternative start site within the holin sequence or occupying an overlapping open reading frame (*Bläsi and Young, 1996*; *Graschopf and Bläsi, 1999*). With these data in mind, we began to investigate Gp138, which we subsequently named ArrA, as a potential antiholin (*Figure 2B*).

The T4 antiholins, RI[T4] and RIII[T4] were initially identified when mutations in these genes demonstrated a rapid lysis plaque morphology. Phages lacking functional antiholins form plaques with sharply defined edges because lysis inhibition no longer occurs. In an effort to find similar rapid-lysing ICP1 mutants with antiholin modifications, we challenged ICP1 with CRISPR-Cas (+) *V. cholerae* containing a spacer targeting *arrA*. We observed a mixture of edge phenotypes including clear-edged plaques (*Figure 3A*) and recovered the fuzzy-edged phenotype by expressing ArrA *in trans* (*Figure 3B*). We engineered a clean Δ*arrA* ICP1 strain to further confirm ArrA function. Of note, and consistent with *arrA* acting as an antiholin, we successfully constructed an *arrA* knockout demonstrating that, despite its conservation, *arrA* is not an essential gene. In support of ArrA acting as an antiholin whose absence results in rapid lysis, Δ*arrA* ICP1 forms plaques that have clear edges (*Figure 3C*).

T4 antiholin mutants demonstrate accelerated lysis kinetics in liquid culture (*Chen and Young, 2016*; *Paddison et al., 1998*), motivating us to test ICP1 Δ*arrA* in liquid cultures. Attempts to obtain high titer stocks of Δ*arrA* ICP1 were unsuccessful (consistent with decreased phage yields) hindering tests of infection at high multiplicities of infection. Instead, we infected PLE (-) *V. cholerae* with Δ*arrA* ICP1, waited for approximately two cycles of infection to complete, and then, once lysis began, we observed a rapid crash in optical density (*Figure 3D*). Such kinetics are in stark contrast to the more prolonged decline seen in wild type infections exhibiting LIN (*Figure 1C*). To be sure this was due to ArrA, we supplied ArrA *in trans* and recovered lysis kinetics characteristic of LIN in which the optical density was stabilized for 30 additional minutes before the culture cleared (*Figure 3D*). To further test that this stabilization of optical density by ArrA when supplied *in trans* was accomplished

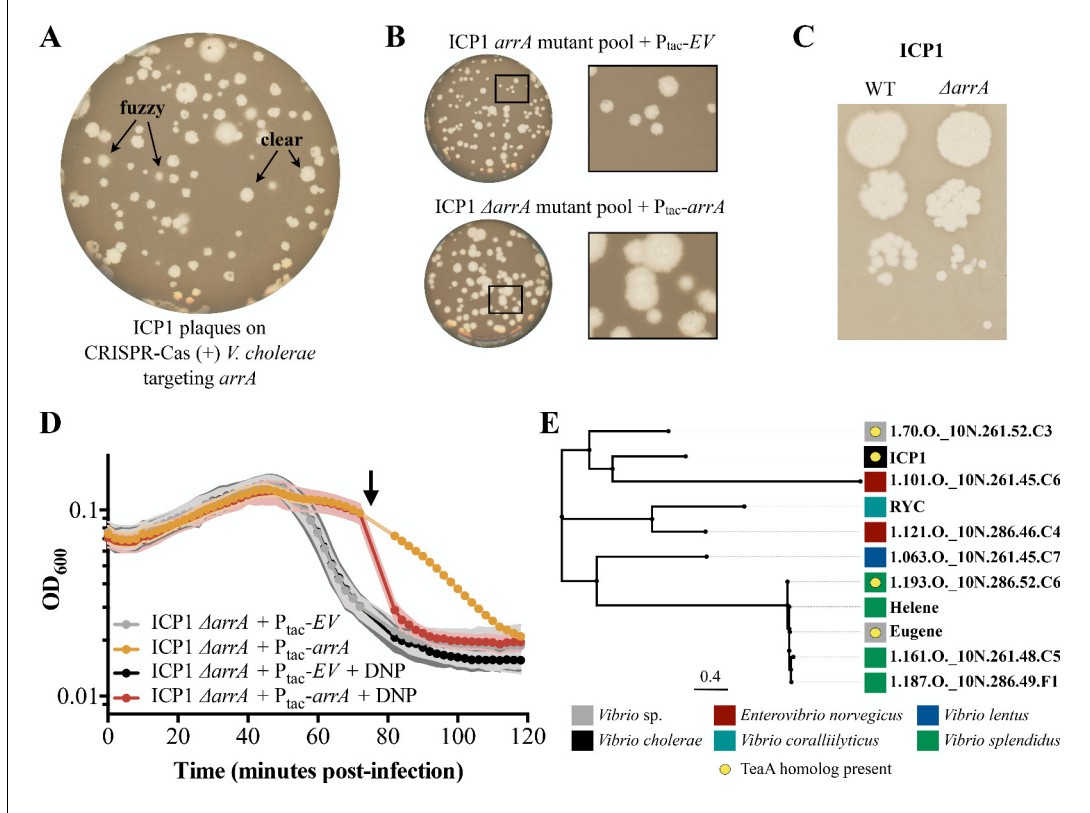

**Figure 3.** Antiholin ArrA identification and characterization. (**A**) Initial plaquing of wild type ICP1 on CRISPR-Cas (+) *V. cholerae* targeting *arrA* yielded a mixture of plaque phenotypes. (**B**) Plaques with clear edges can be found in populations of phage that overcame targeting with various mutations in *arrA* when plaqued on *V. cholerae* harboring an empty vector (*EV*) control (Top). Providing ArrA *in trans* restores the fuzzy edge phenotypes within the mutant population. (**C**) Using a repair template we created a clean *arrA* deletion and found that Δ*arrA* ICP1 yield plaques with clear edges. (**D**) During infection, once lysis begins as observed through changes in optical density ($OD_{600}$), Δ*arrA* ICP1 causes rapid lysis consistent with abolishment of lysis inhibition. When ArrA is expressed *in trans* the lysis inhibition phenotype is rescued: a small lysis event is visible 40 minutes post-infection after which the optical density is stabilized for about 30 minutes. Consistent with ArrA restoring lysis inhibition, the stabilization of optical density is sensitive to chemical disruption of the proton motive force by 2,4-dinitrophenol (DNP) (arrow). Points show the average of four or greater replicates; shading shows the standard deviation. (**E**) BLASTP was used to find proteins with 20% identity to ArrA over 75% of the query; these homologs are displayed in an unrooted tree displaying the name of the phage the protein is found in. Colored blocks show the identity of the host that each phage infects, and yellow circles denote that a protein with homology to TeaA is present in the phage. A table with extended information for each homolog is available in *Supplementary file 3*.

The online version of this article includes the following source data and figure supplement(s) for figure 3:

**Source data 1.** This spreadsheet contains the data used to create *Figure 3D*.

**Figure supplement 1.** Proteins with similarity to TeaA.

through LIN, we exposed cultures to 2,4-dinitrophenol disrupting the proton motive force and collapsing the culture (*Figure 3D*). These data support the conclusion that ArrA is an ICP1-encoded antiholin that helps regulate lysis timing.

BLASTP analysis of TeaA revealed homologs present throughout marine phage and bacterial genomes (*Figure 3—figure supplement 1* and *Supplementary file 2*). ArrA yielded fewer homologs than TeaA (*Supplementary file 3*), however, using less stringent search parameters, we found that some organisms containing TeaA homologs also contain ArrA homologs, though these were limited to vibriophages (*Figure 3E* and *Supplementary file 2 and 3*). This suggests that there are potential homologous LIN systems - complete with both holin and antiholin - present in phages other than ICP1. In contrast, the presence of ArrA homologs without TeaA homologs raises the question: what are antiholins doing on their own? Perhaps they have evolved functionality with holins divergent enough to no longer be considered homologous to TeaA under our search parameters, or they have

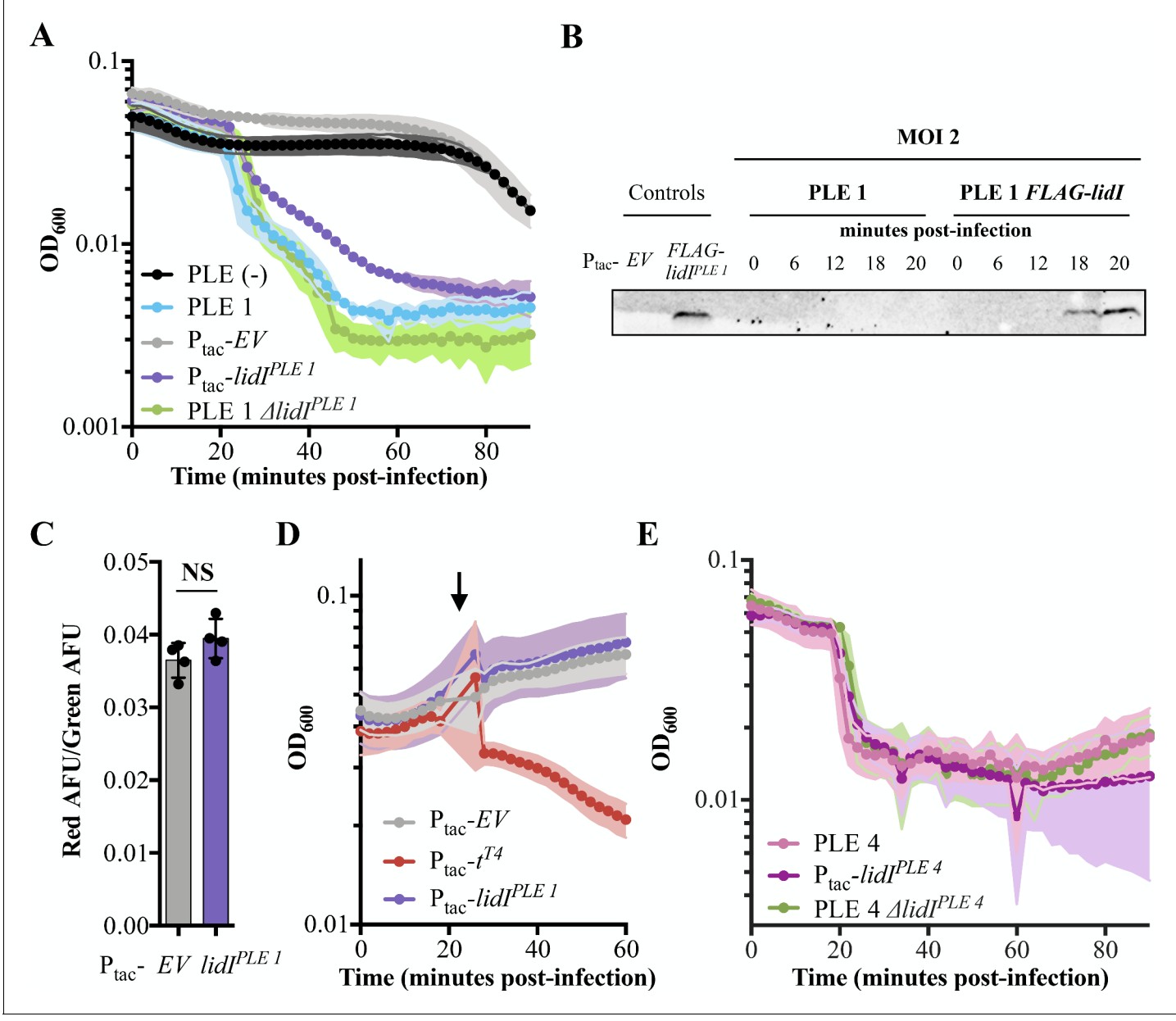

**Figure 4.** Accelerated lysis by PLE and LidI. (**A**) The optical density (OD$_{600}$) of ICP1 MOI = 5 infections was followed in PLE (-), PLE 1, and PLE 1 Δ*lidI V. cholerae* strains as well as *V. cholerae* strains containing induced empty vector (EV) and *lidI*$^{PLE\ 1}$ constructs. Data points represent the average reading of n ≥ 3 replicates and shaded regions display the standard deviation of experiments. The LidI sequence is available in a PRALINE alignment shown in *Figure 4—figure supplement 1*. (**B**) Tagged LidI$^{PLE\ 1}$ expressed *in trans* is readily observable by Western blot. Tagged LidI$^{PLE\ 1}$ in the native PLE context is visible 18 to 20 minutes post infection with ICP1 at MOI = 2. Complete blot available in *Figure 4—figure supplement 2*. (**C**) Proton motive force as measured by the ratio of red to green arbitrary fluorescence units (AFU) from DiOC$_2$(3) after ectopic gene expression. Points represent individual replicates; bar height is the average; error bars display the standard deviation of samples. NS signifies 'no significance' by two-tailed t-test. (**D**) Chemical holin triggering with 2,4-dinitrophenol (arrow) after ectopic gene expression as measured by OD$_{600}$. Points show the average of four replicates; shading shows the standard deviation. (**E**) The OD$_{600}$ of ICP1 at MOI = 5 infections was followed in PLE 4, and PLE 4 Δ*lidI V. cholerae* strains as well as *V. cholerae* containing the induced *lidI*$^{PLE\ 4}$ construct. Data points represent the average reading of n ≥ 5 replicates and shaded regions display the standard deviation of experiments.

The online version of this article includes the following source data and figure supplement(s) for figure 4:

**Source data 1.** This spreadsheet contains the data used to create *Figure 4A and E*.

**Figure supplement 1.** PRALINE alignment of LidI homologs.

**Figure supplement 2.** Complete Western Blot.

**Figure supplement 3.** Redundancy in the PLE.

been coopted for divergent functions much like holins (*Mehner-Breitfeld et al., 2018*; *Saier and Reddy, 2015*).

## PLE accelerates ICP1-mediated lysis

Thus far, characterization of all ICP1 LIN was done in the absence of PLE, a parasitic phage satellite of ICP1. Although the mechanisms that PLE deploys to inhibit and hijack ICP1 are not completely understood, *V. cholerae* lysis kinetics during high multiplicity infections vary depending on the presence of PLE. Consistent with previous experiments at high MOI (MOI = 5) (*O'Hara et al., 2017*), ICP1 infection of PLE (-) *V. cholerae* gradually lyses cultures reaching the lowest optical density ~90 minutes after infection. In contrast, upon infection of PLE 1 *V. cholerae,* rapid lysis starts 20 minutes post-infection (*Figure 4A*). This accelerated timescale could result from any combination of processes such as PLE deploying its own lysis machinery, PLE modulating the expression or stability of ICP1's lysis machinery, or PLE inhibiting or collapsing LIN.

To investigate the underpinnings of accelerated lysis in the presence of PLE, we scrutinized the PLE for potential lysis machinery. Initial analysis revealed no transmembrane domains in any of the ~25 predicted open reading frames in each of PLEs 1, 2, and 3. However, the earliest known PLEs, (PLEs 4 and 5) contain two ORFs with predicted transmembrane domains: ORF2 and ORF26. ORF2$^{PLE\ 4/5}$ does not have homologs in PLEs 1 or 2, suggesting it is not a conserved player mediating accelerated lysis. Consequently, we focused on ORF26$^{PLE\ 4/5}$. Although no homologs were immediately obvious, the synteny between PLEs suggested the presence of previously unannotated open reading frames in PLEs 1, 2, and 3 (namely ORF20.1$^{PLE\ 1}$, ORF24.1$^{PLE\ 2}$ and ORF24.1$^{PLE\ 3}$), which are homologous to ORF26$^{PLE\ 4/5}$ and contain transmembrane domains. We subsequently named these genes *lidI* and the homologs cluster into two groups: *lidI$^{PLE\ 1}$* and *lidI$^{PLE\ 2}$* encode for a 66 amino acid long protein, while *lidI$^{PLE\ 3}$*, *lidI$^{PLE\ 4}$* and *lidI$^{PLE\ 5}$* encode larger proteins at 121 amino acids (*Figure 4—figure supplement 1*). These ORFs have no significant homology to other genes or predicted functional domains beyond their shared transmembrane domains.

Next, to confirm the expression of the newly discovered *lidI* genes, we endogenously tagged LidI-$^{PLE\ 1}$ and evaluated expression during ICP1 infection (*Figure 4B* and *Figure 4—figure supplement 2*). We could not visualize FLAG-LidI$^{PLE\ 1}$ in the absence of phage infection; however, when infected by ICP1 at MOI = 2, FLAG-LidI$^{PLE\ 1}$ was detectable by Western blot late in infection – 18 to 20 minutes post phage addition and immediately prior to the sudden decrease in OD characteristic of PLE-mediated accelerated lysis (*Figure 4B* and *Figure 4—figure supplement 2*).

After confirming *lidI$^{PLE\ 1}$* expression during ICP1 infection, we next sought to characterize LidI$^{PLE\ 1}$ function in PLE (-) *V. cholerae.* During infection with ICP1, LidI$^{PLE\ 1}$ was sufficient to recapitulate the PLE-mediated accelerated lysis phenotype (*Figure 4A*). This phenotype is consistent with *lidI$^{PLE\ 1}$* encoding a holin; however, expression of LidI$^{PLE\ 1}$ at the same level of induction in the absence of phage did not alter cellular proton motive force or make cells susceptible to 2,4-dinitrophenol induced lysis (*Figure 4C and D*). These data suggest that LidI$^{PLE\ 1}$ does not act as a canonical holin when expressed alone and that its ability to mediate cell lysis is dependent on the presence of ICP1.

Having demonstrated that LidI$^{PLE\ 1}$ recapitulates PLE-mediated accelerated lysis, we wanted to determine if it was also necessary for this phenotype. Interestingly, however, when we deleted *lidI$^{PLE\ 1}$* from PLE 1 *V. cholerae* the lysis kinetics were unchanged (*Figure 4A*). To be sure the accelerated lysis is not the consequence of PLE inhibiting the production of phage that are necessary to superinfect cells, exogenous phage was added to PLE 1 and PLE 1 Δ*lidI* strains both of which still demonstrated accelerated lysis (*Figure 4—figure supplement 3A*). As these findings were at odds with the conservation of *lidI* homologs in all the known PLEs, we tested a representative of the other cluster of homologs, LidI$^{PLE\ 4}$, for conserved function. Indeed, LidI$^{PLE\ 4}$ is sufficient to cause accelerated lysis in the absence of PLE, however again we found that it is not necessary – PLE 4 Δ*lidI V. cholerae* strains still exhibit accelerated lysis (*Figure 4E*). To further investigate the role of individual genes in lysis timing, PLE 1 mutants containing a single knockout of each individual open reading frame were exposed to phage and each demonstrated accelerated lysis (*Figure 4—figure supplement 3B*). To identify other PLE-encoded factors sufficient to recapitulate PLE-mediated accelerated lysis, we expressed each PLE 1 ORF individually and challenged those strains with phage. This screen did not identify any other single genes sufficient to accelerate lysis, however, it is important to note that cryptic genes could be responsible, other accelerated lysis systems could require multiple ORFs to function, or the timing and expression level of genes expressed outside the context of PLE could

obscure gene function. Collectively, these results suggest that PLE mediated accelerated cell lysis is the consequence of the activity of two or more functionally redundant gene products, of which LidI is the only product sufficient to phenocopy the PLE-encoded phenotype. Although redundancy is perhaps not expected for mobile genetic elements (MGEs) with restricted genome size, we have additionally observed that no single PLE open reading frame is necessary for inhibition of ICP1 plaque formation (*Figure 4—figure supplement 3C*), suggesting that multiple strategies act synergistically to eliminate phage production in addition to accelerating lysis.

## LidI[PLE 1] can function through lysis inhibition disruption

As LidI[PLE 1] is sufficient to accelerate lysis but does not phenocopy what we expect of a holin, we wanted to determine the mechanism of LidI[PLE 1]-mediated accelerated lysis in the absence of PLE. Since ICP1 exhibits LIN during high MOI infections, we hypothesized that LidI[PLE 1] causes accelerated lysis by disrupting LIN. LIN occurs when phages outnumber hosts, so changing the multiplicity of infection changes the onset of LIN: at low multiplicities of infection, a small fraction of cells produce a burst of phage which are adsorbed by neighbors and this process can repeat a number of times until the majority of cells are infected and LIN is triggered (*Figure 1A*). This is evidenced by the stabilization of culture optical density until complete lysis ~90 minutes in PLE (-) *V. cholerae* infected at various multiplicities of infection (*Figure 5A*). The differential onset of LIN means that accelerated lysis mediated by LidI[PLE 1] would also be expected to change in accordance with MOI if it functions by disrupting LIN. Indeed, we observed differential lysis timing dependent on the MOI in cultures expressing LidI[PLE 1] with up to a 40 minutes delay at the lowest MOI (*Figure 5A*), suggesting that LidI[PLE 1] functions through lysis inhibition disruption.

Congruent with LidI[PLE 1] disrupting LIN, *lidI[PLE 1]* expression in PLE (-) *V. cholerae* does not change the efficiency of plaquing (EOP) by ICP1, an experiment that probes the number of successful initial infections at a low multiplicity of infection (*Figure 5B*). Consistent with this, the phenotypic change in plaque morphology expected of disrupted LIN is the loss of fuzzy plaque edges, which we see in PLE (-) *V. cholerae* expressing *lidI[PLE 1]* in trans (*Figure 5C*). It is important to note that these data showing that LidI disrupts LIN when expressed alone do not reveal the molecular mechanism underlying this activity or ensure that the gene serves the same function in the context of PLE, even though it successfully phenocopies PLE-induced accelerated lysis.

## LidI lowers phage yield

Because LIN in T-even coliphages functions to increase phage burst size (*Doermann, 1948*), we hypothesized that LidI[PLE 1] collapsing LIN could inhibit ICP1 by decreasing progeny phage yield from infection. To determine if LidI[PLE 1] alone can impact the number of phage produced from an infection, expression was induced prior to, at the time of, and at various intervals after infection with ICP1 at a high MOI (MOI = 5). Induction of *lidI[PLE 1]* at the time of infection or 20 minutes before infection resulted in decreased phage yield by one or two orders of magnitude, respectively, in comparison to strains with the empty vector control (*Figure 6A*). Not only did we find this to be true of LidI[PLE 1], but we also tested the LidI[PLE 4] homolog for conserved inhibition of ICP1 and observed the same decreased phage yield (*Figure 6B*). Subsequent testing revealed that when infections start out with a low number of phage per cell (MOI ≤0.001), there is still accelerated lysis and decreased progeny phage production when LidI[PLE 1] expression is induced (*Figure 6—figure supplement 1*). These data reveal LidI as the first PLE-encoded ORF that can singlehandedly negatively impact ICP1 phage yield.

From an evolutionary perspective, producing fewer virions equates to fewer diverse phages and would limit the phage's ability to evolve counterattacks to anti-phage mechanisms or escape through mutation. Hence, we hypothesize that PLE-mediated accelerated lysis decreases the ability of ICP1 to evolve in the face of PLE. However, because our data support a model in which accelerated lysis is redundantly encoded, it is not currently possible to test the impact of delayed lysis on ICP1 evolution in the context of the PLE. We can, however, interrogate how the *lidI*-mediated collapse of LIN and concomitant decrease in phage production in PLE (-) *V. cholerae* constrains ICP1 evolution. To test if LidI[PLE 1]-mediated accelerated lysis is enough to impact diversity through the acquisition of random mutations in the progeny phage population, we exposed PLE (-) *V. cholerae* with and without *lidI[PLE 1]* to ICP1 (MOI = 0.1), collected the population of progeny phage, and

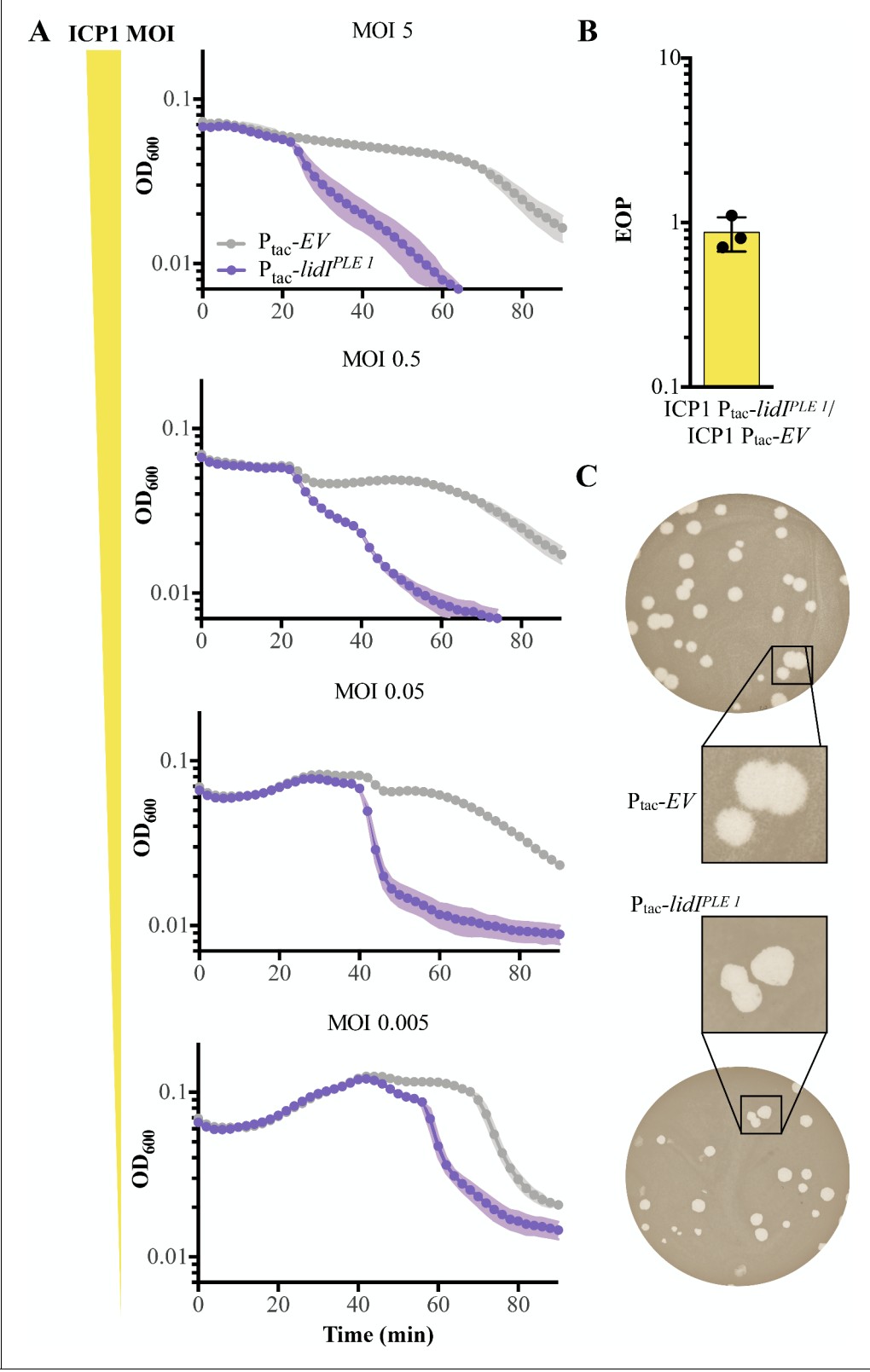

**Figure 5.** LidI functions through lysis inhibition disruption. (**A**) Optical density ($D_{600}$) of empty vector (EV) or LidI[PLE 1] expressing cultures after infection with different initial multiplicities of infection (highest MOI top to lowest MOI bottom). Data points represent the average reading of n = 3 biological replicates and shaded regions display the standard deviation of experiments. (**B**) Efficiency of plaquing (EOP) for *V. cholerae* harboring induced *lidI*[PLE 1]

*Figure 5 continued on next page*

Figure 5 continued

in comparison to an empty vector control (EV). Points represent individual replicates; bar height is the average; error bars display the standard deviation of samples. (C) Plaque edge phenotypes were determined for ICP1 plaques on EV (top) and *lidl*^PLE 1 (bottom) PLE (-) *V. cholerae*.

The online version of this article includes the following source data for figure 5:

**Source data 1.** This spreadsheet contains the data used to create *Figure 5*.

looked for plaque formation on PLE (-) *V. cholerae* encoding a Type I-E CRISPR-Cas system (*Box et al., 2016*); an expanded schematic of this experiment is available in *Figure 6—figure supplement 2*. These host strains of CRISPR-Cas (+) *V. cholerae* were engineered to harbor various anti-ICP1 spacers that allowed for varying rates of ICP1 escape (*Figure 6C*). For each spacer, fewer phage progeny from *lidl*^PLE 1 *V. cholerae* were able to overcome targeting than progeny from infections of strains without *lidl*^PLE 1 (*Figure 6D*). This defect is due to the LidI-mediated decrease in the population of phages as the frequency of phage escaping stays the same (e.g. ~2 out of every thousand phages can overcome spacer C, *Figure 6D*). Consequently, because less phage are produced from *lidl*^PLE 1-expressing *V. cholerae,* an order of magnitude fewer phages can overcome the spacer in the population (*Figure 6—figure supplement 4*).

While exposure to *V. cholerae*'s CRISPR-Cas system was meant to probe evolution at the level of individual random mutations and proclivity to overcome targeting, phages also readily use homologous recombination to evolve during co-infection. To test the impact of LidI^PLE 1 on this aspect of evolvability, we took advantage of the Type I-F CRISPR-Cas system found in ICP1 by engineering two ICP1 variants with nonfunctional CRISPR-Cas systems: one devoid of spacers against PLE with an inactive Cas1 preventing spacer acquisition (CRISPR*-Cas ICP1) (*McKitterick et al., 2019b*) and the other lacking Cas2-3 (CRISPR-Cas* ICP1). We then used these phages to coinfect *V. cholerae* strains (MOI = 0.01) with and without LidI^PLE 1. After lysis, progeny phage were tested for their ability to plaque on PLE 1 *V. cholerae,* which is only possible if homologous recombination between the two variants restored a functional CRISPR-Cas system able to target PLE 1 (*Figure 6E*); an expanded schematic of this experiment is available in *Figure 6—figure supplement 3*. Predictably, the presence of LidI^PLE 1 decreased the number of progeny phages per infection that recombined to reconstitute the CRISPR-Cas system and overcome PLE (*Figure 6F* and *Figure 6—figure supplement 4*). Although LidI expression does not directly impact ICP1 evolvability within infected cells, these data demonstrate that LidI imposes a bottleneck on ICP1's population size resulting in fewer diverse phages and limiting ICP1's potential to escape anti-phage activity.

## Discussion

Here, we observe a previously unknown lysis inhibition (LIN) state in the globally relevant pathogen, *V. cholerae,* in response to ICP1, the predominant phage isolated from clinical samples, and identify the relevant ICP1-encoded lysis machinery: the holin, TeaA, and the antiholin, ArrA. Consistent with T-even induced LIN, ICP1 LIN results in fuzzy-edged plaques and delayed lysis sensitive to cellular proton motive force. This work subsequently reveals that the disruption of LIN is part of PLE's anti-phage repertoire. A single open reading frame, *lidl*, which is conserved through all five PLEs spanning the last 70 years, is sufficient to disrupt LIN and limit progeny phage populations when expressed outside its native context in PLE (-) *V. cholerae*. These two opposing forces, ICP1 LIN and accelerated lysis by PLE through LIN disruption (which our data shows LidI is capable of doing in isolation and yet other undiscovered PLE-encoded mechanisms redundantly accomplish), act in the midst of the ongoing evolutionary arms race between *V. cholerae* and its parasites.

In phage-host interactions, research continues to unveil evolutionary strategies used to sense populations in the environment. Phage-encoded anti-CRISPRs (the counter adaption to combat CRISPR-Cas systems) can function in a phage-concentration dependent manner (*Borges et al., 2018*; *Landsberger et al., 2018*), small arbitrium peptides can influence lysogeny decisions based on other infections throughout the population (*Erez et al., 2017*), and phages can 'listen in' on host quorum sensing as a measure of host availability (*Silpe and Bassler, 2019*). All of these environmental signals parallel LIN, one of the first discovered forms of communication during phage infection (*Doermann, 1948*; *Abedon, 2019*; *Hershey, 1946*). The anti-CRISPR system uses multiple subsequent phage infections, each benefiting from anti-CRISPRs expended during previous infections, to

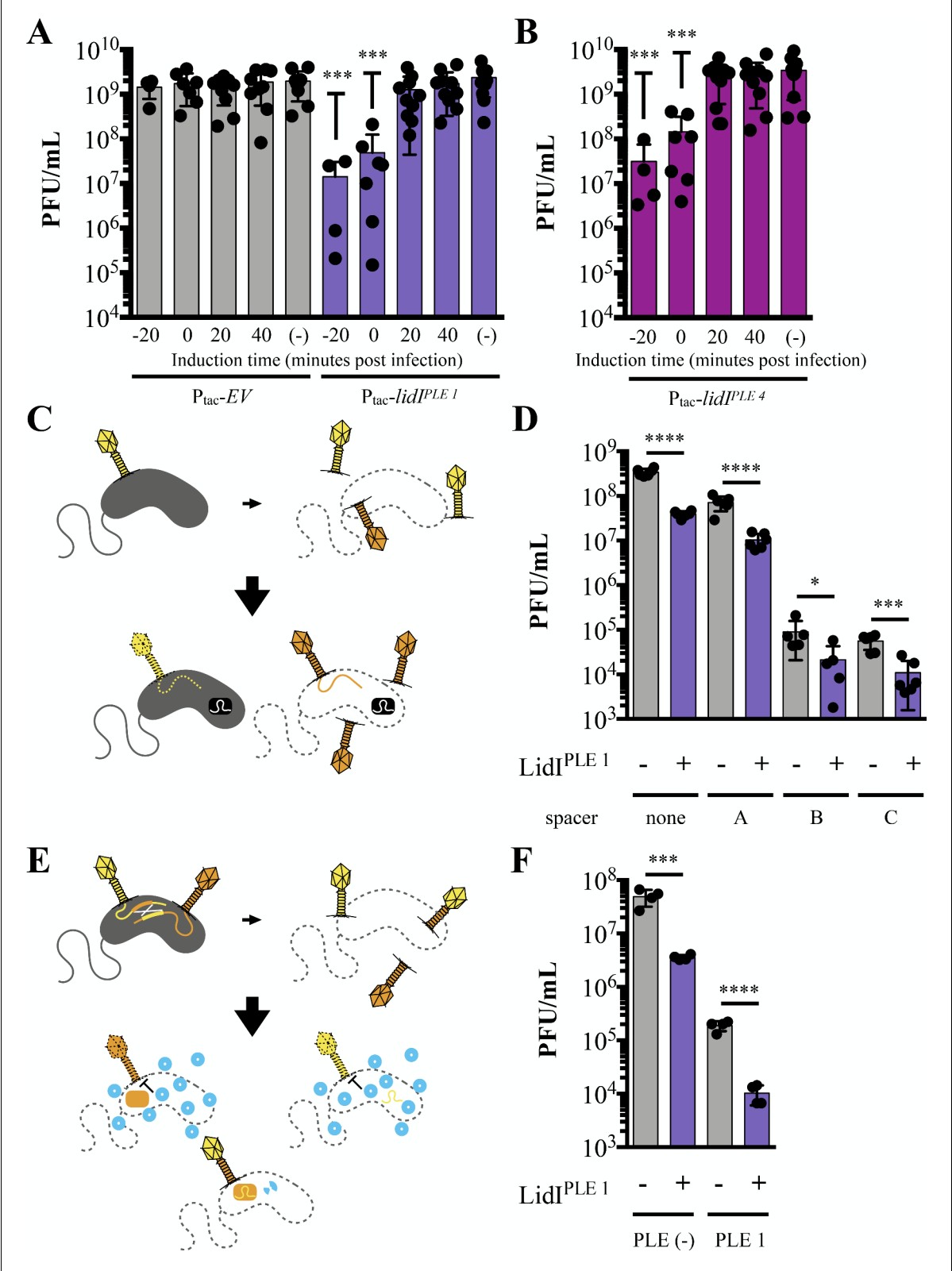

**Figure 6.** LidI puts a bottleneck on phage populations. (**A and B**) Phage infection yields measured in plaque forming units per mL (PFU/mL) were determined from *V. cholerae* with empty vector (EV), *lidI*<sup>PLE 1</sup>, and *lidI*<sup>PLE 4</sup> constructs induced at the specified time with respect to phage infection at multiplicity of infection of 5 or left uninduced (-). All samples were mechanically lysed 100 minutes after infection and titers were compared via one-way ANOVA followed by Dunnett's multiple comparison test comparing all values to titers from the EV cultures. Only significant differences are noted. (**C**)
*Figure 6 continued on next page*

*Figure 6 continued*

Schematic of evolution as probed by overcoming *V. cholerae*-encoded CRISPR targeting. Top: cultures with induced EV and *lidI*[PLE 1] constructs are infected with ICP1 (MOI = 0.1) and allowed to produce progeny phage before mechanical lysis. The majority of phages will be wild type (yellow) while some phages will harbor mutations (orange). Bottom: The phage populations were then plaqued on *V. cholerae* strains with a CRISPR-Cas system (black rectangle) and different spacers. Wild type phage infections do not result in plaques because the injected phage DNA is degraded (dotted line) while mutants that can overcome the targeting, produce progeny, lyse the cells and form plaques. Expanded schematic in *Figure 6—figure supplement 2*. (D) Quantification of the number of phages that could overcome targeting by CRISPR-Cas from populations of progeny phages with and without LidI[PLE 1]. Significance was determined by unpaired one-tailed t-tests between EV and *lidI*-expressing hosts. (E) Schematic of evolution by successful homologous recombination between coinfecting phages. Top: cultures with induced EV and *lidI*[PLE 1]constructs are coinfected with ICP1 harboring mutated, non-functional CRISPR-Cas systems: CRISPR*-Cas ICP1 (orange) and CRISPR-Cas* ICP1 (yellow) at MOI 0.01. Bottom: The progeny phage from these infections were then plated on PLE 1 *V. cholerae*. PLE (blue circles) inhibit phage production by all phages that did not successfully recombine to restore a functional CRISPR-Cas system (two-colored phage). Expanded schematic in *Figure 6—figure supplement 3*. (F) Progeny phage from hosts with and without LidI[PLE 1] co-infected with CRISPR*-Cas ICP1 and CRISPR-Cas* ICP1 were harvested via mechanical lysis. These were then plaqued on permissive (PLE (-) *V. cholerae*) and restrictive (PLE 1 *V. cholerae*) hosts. Significance was determined via one-tailed t-tests between EV and *lidI*-expressing hosts. For all graphs, data points represent individual values, bar height represents the average value, and error bars represent the standard deviation. Additional efficiency of plaquing analysis of 6D and 6F can be found in *Figure 6—figure supplement 4*. *p≤0.05, **p≤0.01, ***p≤0.001, ****p≤0.0001.

The online version of this article includes the following source data and figure supplement(s) for figure 6:

**Source data 1.** This spreadsheet contains the data used to create *Figure 6*.
**Figure supplement 1.** Accelerated lysis and phage yield after low MOI infections.
**Figure supplement 1—source data 1.** This spreadsheet contains the data used to create *Figure 6—figure supplement 1*.
**Figure supplement 2.** Diversity of progeny phage from infections as measured by phage escape from CRISPR-Cas – expanded schematic.
**Figure supplement 3.** Successful recombination of progeny phage from infections as measured by phage-encoded CRISPR-Cas overcoming PLE – expanded schematic.
**Figure supplement 4.** EOP of evolution experiments in the presence of LidI[PLE 1].
**Figure supplement 4—source data 1.** This spreadsheet contains the data used to create *Figure 6—figure supplement 4*.

overcome host defenses similar to the subsequent superinfections that trigger LIN to make the most of the infected host. Arbitrium peptide signals formed upon infection give phages a proxy for when the majority of neighboring cells are infected, protecting potential progeny phage from adsorbing to previously infected cells by employing lysogeny – this shielding of progeny phages from infected cells is also accomplished during LIN. Phages eaves dropping on the host quorum sensing system provides a measure of available hosts – a lack of which is communicated by secondarily adsorbed virions during LIN. Consequently, our discovery of ICP1's LIN in *V. cholerae* reinforces lysis inhibition as a relevant form of environmental sensing and highlights the importance of phages tuning their infection parameters depending on host availability in the environment. Following ingestion, *V. cholerae* colonizes and blooms in the small intestine before being shed in stool, further contaminating aquatic reservoirs and promoting subsequent ingestion and infection cycles. If large numbers of ICP1 are co-ingested with small numbers of *V. cholerae,* it would be theoretically beneficial for ICP1 to demonstrate LIN in infected cells, bide time inside the cell while making more virions, and lyse in the small intestine after uninfected *V. cholerae* have had time to replicate, providing ample hosts for subsequent rounds of phage predation. This is consistent with outbreak models in which phage become more abundant in the environment towards the end of an epidemic and there is phage amplification within cholera patients (*Faruque et al., 2005a*; *Faruque et al., 2005b*; *Jensen et al., 2006*). Similarly, if *V. cholerae* bloomed in the gut followed by a subsequent expansion of the phage population to the extent that phages now outnumber hosts, it would be beneficial for ICP1 progeny phage to stay inside a cell, protected from adsorbing to cells that are already making progeny phage, and instead wait for release into the aquatic environment or ingestion by an uninfected patient via person-to-person transmission where ICP1 progeny may have better chances of finding an uninfected bacterial host to carry out its parasitic lifecycle. Interestingly, this begins to touch on variability of the phage life cycle in different aquatic environments (*Nelson et al., 2008*; *Silva-Valenzuela and Camilli, 2019*), and the potential benefit for a phage population if a lysis inhibited cell is ingested along with *V. cholerae* from a patient – after which *V. cholerae* has been shown to be hyperinfectious (*Merrell et al., 2002*).

It is important to consider that within this tripartite system there are two parasites at odds with one another and both encode mechanisms to alter lysis timing. While the benefits of LIN to ICP1

have been explored in the parallel T-even phage/*E. coli* systems (*Abedon, 1990*; *Abedon, 2019*) and discussed above, the impact of a MGE, specifically a parasitic phage satellite like PLE, on lysis inhibition could not have been predicted. One could argue that LIN might provide PLE with more resources for horizontal transmission, or counter that LIN provides time enough for ICP1 to evade PLE-encoded anti-phage mechanisms. Here, we show that the PLE accelerates lysis with LidI as a conserved part of its program – presumably collapsing LIN as a hinderance to ICP1. Known examples showcase the deleterious effects of accelerated lysis on phage fitness, with holin mutations like the λ phage S-holin mutants which accelerate lysis by 20–25 minutes and lower progeny phage yield by orders of magnitude (*Johnson-Boaz et al., 1994*; *Wang, 2006*). Similarly, a part of an abortive infection system in *Lactococcus lactis,* the AbiZ protein, causes cells infected by φ31 to lyse 15 minutes early, decreasing phage titers 100-fold (*Durmaz and Klaenhammer, 2007*). Indeed, we find even without other PLE-encoded products, accelerated cell lysis by LidI is sufficient to decrease phage population size and this bottlenecks the phage population, likely reducing ICP1's ability to overcome PLE. The absolute abolishment of progeny phage accomplished by PLE's complete anti-phage repertoire (of which LidI is only a part) is particularly interesting considering functionally similar SaPIs, which lay dormant in the chromosome much like PLE, until phage infection when SaPIs are induced to parasitize phage components (*Novick et al., 2010*). One of many ways that PLEs and SaPIs differ is that SaPIs allow for some propagation of their helper phage, whereas PLE completely ablates ICP1 production of progeny phage. It is easy to think of SaPIs as selfish elements: they integrate and take advantage of vertical transmission. Once the cells are challenged by a helper phage, they excise, inhibit phage for their own ends, and escape the cell while allowing some progeny phage to escape with them all the while promoting diversity and horizontal gene transfer (*Frígols et al., 2015*; *Novick et al., 2010*). This lifecycle ensures horizontal transmission of the SaPI as well as continued activation of SaPIs down the line by available helper phages. In contrast, PLE completely blocks ICP1 production by acting as an abortive infection mechanism. Our evidence that PLE functions through collapsing lysis inhibition supports this angle as lysis inhibition could theoretically, as previously mentioned, allow for more time to produce PLE particles, enabling horizontal transmission of PLE to larger numbers of naïve cells. Surprisingly, this is so important that the means to disrupt LIN and execute lysis for PLE's own selfish benefit are apparently redundantly encoded within the PLE – LidI can collapse LIN in the absence of PLE, though Δ*lidI* PLE still shows accelerated lysis. There is also limited evidence that PLE uses the hijacked ICP1 machinery to transduce in nature – in the laboratory, conditions allow four of the five PLEs to integrate in many sites across the superintegron; however, natural isolates only ever have one of those four PLEs integrated in one specific site (*O'Hara et al., 2017*). This pattern is indicative of vertical transmission and infrequent horizontal transduction in the strains sampled from epidemics, which makes it easier to reconcile PLE's abortive infection activity. With these evolutionary hypotheses in mind, ICP1 acquiring the CRISPR-Cas system changed the game: some single spacers encoded in ICP1 targeting PLE allow ICP1 to form progeny while simultaneously allowing for transduction of PLE (*McKitterick et al., 2019b*). If all spacers were singular and created equal, selection could drive PLE to act more like a typical satellite phage, embracing horizontal transfer and allowing ICP1 to slide by producing limited progeny phage. We know, however, that this is not the case; CRISPR systems function by dynamically acquiring spacers (*Barrangou et al., 2007*) and multiple spacers can abolish PLE's ability to transduce while also killing the cell harboring PLE, destroying any chance at horizontal or vertical transmission (*McKitterick et al., 2019b*). Considering this complication, it is no surprise PLE would employ products like LidI to collapse LIN, perhaps to limit the ability of ICP1 to pick up new spacers against PLE.

The dynamic arms race between ICP1, PLE, and *V. cholerae* is ongoing as is research on other coevolving parasite/host systems. Focusing future work on LIN and MGEs is particularly promising given that this work represents a novel incarnation of LIN outside of the T-even coliphages, and we found homologs of LIN machinery outside of the limited contexts that LIN has been previously alluded to (*Gromkova, 1968*; *Latino et al., 2019*; *Schito, 1974*). This prevalence suggests that LIN exists outside of characterized systems though the impacts of LIN and its disruption are unknown and largely unexplored. In contrast, the importance of MGEs is widely accepted and anti-phage mechanisms are increasingly found on MGEs, making questions about the interplay between MGEs and the complicating factors outlined here particularly attractive. One recently discovered example of how prevalent these confounding factors are is that of the cyclic-oligonucleotide-based anti-phage signaling system, which is found on MGEs in bacteria including *V. cholerae*, and that, in *E.*

*coli*, serves as an abortive infection system upon phage infection and can cause lysis on an accelerated timescale (*Cohen et al., 2019*). As a final note on these intriguing areas of future inquiry, there is increased interest in utilizing phages to combat bacterial infections as a part of phage therapy - the successful application of such approaches will likely depend on understanding all the interactions between phages and bacteria including responses that depend on the environment like lysis inhibition and interplay mediated by MGEs like PLE.

# Materials and methods

## Key resources table

| Reagent type (species) or resource | Designation | Source or reference | Identifiers | Additional information |
|---|---|---|---|---|
| Gene (*Vibrio cholerae*) | *lidI*$^{PLE\ 1}$ (PLE 1 ORF20.1) | * | | |
| Gene (*Vibrio cholerae*) | *lidI*$^{PLE\ 2}$ (PLE 2 ORF24.1) | * | | |
| Gene (*Vibrio cholerae*) | *lidI*$^{PLE\ 3}$ (PLE 3 ORF24.1) | * | | |
| Gene (*Vibrio cholerae*) | *lidI*$^{PLE\ 4}$ (PLE 4 ORF26) | (*O'Hara et al., 2017*) | | |
| Gene (*Vibrio cholerae*) | *lidI*$^{PLE\ 5}$ (PLE 5 ORF26) | (*O'Hara et al., 2017*) | | |
| Gene (bacteriophage ICP1) | *teaA*$^{ICP1}$ (gp137) | (*Angermeyer et al., 2018*),* | | |
| Gene (bacteriophage ICP1) | *arrA*$^{ICP1}$ (gp138) | (*Angermeyer et al., 2018*),* | | |
| Strain (*Vibrio cholerae*) | PLE (-) *V. cholerae* (E7946) | (*Levine et al., 1982*) | KDS 6 | |
| Strain (*Vibrio cholerae*) | PLE 1 *V. cholerae* (PLE 1 E7946) | (*O'Hara et al., 2017*) | KDS 36 | |
| Strain (*Vibrio cholerae*) | Δ*lacZ*::P$_{tac}$-*EV* (E7946) | (*McKitterick and Seed, 2018*) | KDS 116 | |
| Strain (*Vibrio cholerae*) | Δ*lacZ*::P$_{tac}$-*lidI*$^{PLE\ 1}$ (E7946) | * | KDS 139 | |
| Strain (*Vibrio cholerae*) | Δ*lacZ*::P$_{tac}$-*lidI*$^{PLE\ 4}$ (E7946) | * | KDS 267 | |
| Strain (*Vibrio cholerae*) | PLE 1 FLAG-LidI$^{PLE\ 1}$ (E7946) | * | KDS 268 | |
| Strain (*Vibrio cholerae*) | PLE 1 Δ*lidI* (E7946) | * | KDS 170 | |
| Strain (*Vibrio cholerae*) | PLE 4 Δ*lidI* (E7946) | * | KDS 269 | |
| Strain (*Vibrio cholerae*) | CRISPR-Cas (+) (E7946) | (*Box et al., 2016*) | KDS 112 | Inducible Cas Proteins |
| Recombinant DNA reagent (plasmid) | P$_{tac}$-*Empty Vector* (pKL06 in E7946) | (*McKitterick and Seed, 2018*) | KDS 196 | Empty Vector Control |
| Recombinant DNA reagent (plasmid) | P$_{tac}$-*lidI*$^{PLE\ 1}$ (plasmid in E7946) | * | KDS 219 | Inducible *lidI*$^{PLE\ 1}$ |
| Recombinant DNA reagent (plasmid) | P$_{tac}$-*lidI*$^{PLE\ 4}$ (plasmid in E7946) | * | KDS 270 | Inducible *lidI*$^{PLE\ 4}$ |
| Recombinant DNA reagent (plasmid) | P$_{tac}$-*teaA*$^{ICP1}$ (plasmid in E7946) | * | KDS 271 | Inducible *teaA*$^{ICP1}$ |
| Recombinant DNA reagent (plasmid) | P$_{tac}$-*arrA*$^{ICP1}$ (plasmid in E7946) | * | KDS 272 | Inducible *arrA*$^{ICP1}$ |
| Recombinant DNA reagent (plasmid) | P$_{tac}$-*t*$^{T4}$ (plasmid in E7946) | * | KDS 273 | Inducible *t*$^{T4}$ |
| Recombinant DNA reagent (plasmid) | P$_{tac}$-*anti-gp138 spacer* (plasmid in E7946) | * | KDS 274 | CRISPR array containing anti-gp138 spacer |

*Continued on next page*

*Continued*

| Reagent type (species) or resource | Designation | Source or reference | Identifiers | Additional information |
|---|---|---|---|---|
| Recombinant DNA reagent (plasmid) | $P_{tac}$-*anti-gp138 spacer* and repair template | * | KDS 275 | CRISPR array containing anti-gp138 spacer and repair template |
| Recombinant DNA reagent (plasmid) | $P_{tac}$-*FLAG-lidl*[PLE 1] | * | KDS 276 | Inducible FLAG-tagged blot control |
| Recombinant DNA reagent (plasmid) | $P_{tac}$-*none* (CRISPR array with no spacers against WT ICP1) | (*McKitterick et al., 2019b*) | KDS 277 | Spacer control |
| Recombinant DNA reagent (plasmid) | $P_{tac}$-*spacer A* | * | KDS 278 | Spacer A against ICP1 |
| Recombinant DNA reagent (plasmid) | $P_{tac}$-*spacer B* | * | KDS 279 | Spacer B against ICP1 |
| Recombinant DNA reagent (plasmid) | $P_{tac}$-*spacer C* | * | KDS 280 | Spacer C against ICP1 |
| Strain (bacteriophage ICP1) | ICP1 (ICP1 2006E ΔCRISPR ΔCas) | (*McKitterick and Seed, 2018*) | | |
| Strain (bacteriophage ICP1) | Δ*arrA* ICP1 (ICP1 2006E Δ*arrA/gp137*) | * | SGH Φ 61 | |
| Strain (bacteriophage ICP1) | CRISPR*-Cas ICP1 (ICP1 2011A Δ*spacer2-9* Cas1D244A) | (*McKitterick et al., 2019b*) | ACM Φ 232 | |
| Strain (bacteriophage ICP1) | CRISPR-Cas* ICP1 (ICP1 2011A Δ*cas2-3*) | * | SGH Φ 62 | |
| Chemical compound | Isopropyl-beta-D-thiogalactoside (IPTG) | GoldBio | 12481C5 | |
| Chemical compound | Theophylline | Sigma-Aldrich | T1633-100G | |
| Chemical compound | 2,4-Dinitrophenol (DNP) | Sigma-Aldrich | D198501-100G | |
| Chemical compound | 3,3-Diethyloxacarbo cyanine iodide ($DiOC_2(3)$) | Sigma-Aldrich | 320684–1G | |
| Antibody | Rabbit anti-FLAG polyclonal antibody | Sigma-Aldrich | RRID:SAB4301135 | |
| Antibody | Goat anti-Rabbit IgG antibody, peroxidase conjugated | Sigma-Aldrich | RRID:AP132P | |

*Identified or created in this work.

## Bacteria and phage propagation

Bacteria were propagated at 37°C via streaking from frozen glycerol stocks on solid LB agar plates and growth in Miller LB (Fisher Bioreagents) with aeration. Media was supplemented with chloramphenicol (2.5 µg/mL for *V. cholerae* and 25 µg/mL chloramphenicol for *E. coli*), kanamycin (75 µg/mL), ampicillin (100 µg/mL), and streptomycin (100 µg/mL) when appropriate. Cell densities were measured at $OD_{600}$ in tubes (Biochrom Ultrospec 10; 10 mm pathlength referred to as $OD_{600-tube}$) and in 96-well plates (all reported $OD_{600}$ measurements in figures have a pathlength equivalent to 150 µL in Costar Clear 96-well plates (Corning)). To induce chromosomal constructs in both liquid and top agar, plasmid constructs in top agar, and the plasmid $P_{tac}$-*arrA* construct for complementation, 1 mM isopropyl β-D-1-thiogalactopyranoside (IPTG) and 1.5 mM theophylline were added to cultures while the remaining plasmid constructs were induced with 125 µM IPTG and 187.5 µM theophylline in liquid cultures. Plasmid constructs were used for all experiments other than phage infection yield experiments which utilized chromosomal constructs to decrease leaky expression in uninduced strains.

Bacteriophages were propagated on PLE (-) *V. cholerae* hosts and prepped via polyethylene glycerol precipitation or concentration and media exchange on Amicon Ultra – 15 (Millipore) centrifugal filters (*Bonilla et al., 2016*; *Clokie and Kropinski, 2009*). Stocks were stored in sodium chloride-tris-ethylenediaminetetraacetic acid buffer (STE), and quantified via the soft agar overlay method (*Clokie and Kropinski, 2009*). Briefly, titering was completed by growing *V. cholerae* to mid-log, infecting with cultures with diluted phage, and allowing adsorption to occur for 7 to 10 minutes before plating on 0.5% LB top agar. Subsequently, multiplicity of infection (MOI) was determined by calculating the number of plaque forming units and varying that with the number of colony forming units of *V. cholerae* at a given optical density. This does not take into account virions that adsorb but do not successfully form plaques. Consequently, all reported MOIs do not address multiplicity of adsorption which could vary and impact initial changes in optical densities during experiments. Mechanical lysis of cultures infected with phage was accomplished by mixing chloroform into cultures then letting cultures stand at room temperature for ten minutes before spinning at 5000 x *g* for 15 minutes at 4°C and removing the supernatant for further analysis.

## Cloning and strain generation

Chromosomal integrations in *V. cholerae* were accomplished through natural transformation of linear DNA created via splicing by overlap extension PCR (*Dalia et al., 2014*). In the case of deletions, antibiotic resistance cassettes were integrated into the locus of the deleted gene and subsequently flipped out as previously described (*Baba et al., 2006*). Plasmids were constructed with Gibson Assembly and Golden Gate reactions. Phage mutants were selected as previously described (*Box et al., 2016*). Briefly, complementary spacer oligos were annealed and inserted into a plasmid-borne CRISPR array. This plasmid was mated into a strain of *V. cholerae* engineered to include an inducible Type 1-E CRISPR-Cas system (CRISPR-Cas (+) *V. cholerae*). This system in the host strain was induced for 20 minutes before ICP1 infection for plaque assays on 0.5% LB top agar containing antibiotics to maintain the CRISPR array plasmid. Plaques were picked into STE (100 mM NaCl, 10 mM Tris-HCl 1 mM EDTA), purified on the same host twice, and genomic DNA was prepped for PCRs with the DNeasy Blood and Tissue Kit (Qiagen). Phages were subjected to PCR of the targeted gene and subsequent Sanger sequencing. Clean knockouts were accomplished by adding a repair template of homologous sequence containing the desired deletion and flanking DNA to the plasmid containing the CRISPR array as previously described (*Box et al., 2016*).

## Lysis kinetics

*V. cholerae* strains were grown in 2 mL cultures to an $OD_{600-tube}=0.3$ and 150 µL of cultures were added to 96-well plates. This transition often results in a slight decrease in optical density at the beginning of experiments where $OD_{600}$ is tracked. The underlying cause for this decrease is unknown, but is consistent between controls and experimental conditions. Inducers and phage were pre-aliquoted in plates unless otherwise specified. $OD_{600}$ within the plate was read for each sample on the SpectraMax i3x (Molecular Devices) plate reader every two minutes with one minute of shaking between each read while the machine incubated cultures at 37°C. Assays were interrupted for the addition of phage, inducers, 2,4-dinitrophenol (DNP; 2 µL of 8.4 mM DNP dissolved in 80% ethanol for a final concentration of 110 µM DNP), or ethanol (2 µL of 80% ethanol) during which samples were removed from the plate reader briefly before measurement was resumed. This enabled superinfection of cultures (initial cultures were infected with ICP1 MOI = 1, returned to the plate reader for four minutes, and subsequently superinfected with ICP1 MOSI = 5), addition of DNP in ethanol or ethanol alone to ICP1 MOI = 5 infected cultures 25 minutes post-infection, and DNP addition to induced cultures after 20 minuntes of growth with the inducers.

## Determination of phage yield from high MOI infection

For each strain, three 2 mL cultures of *V. cholerae* were grown. The first was grown to an $OD_{600-tube}=0.15$, inducer was added, and the culture was returned to the incubator; this culture served as the pre-induced culture. All cultures were grown for an additional 20 minutes to $OD_{600-tube}=0.3$ at which point ICP1 MOI = 5 was added. Inducer was added to one tube at this time; this culture served as the culture induced at time zero. Tubes were returned to the incubator for 5 minutes for phage adsorption. Cultures were spun at 5000 x *g* for 3 minutes to pellet cells. Unadsorbed phage

was aspirated off and cells were washed once with 1 mL of prewarmed LB with or without inducer. Cells were spun again and resuspended in media with and without inducer at which point $OD_{600-tube}$ was determined. Cultures (150 µL) were moved to 96-well plates with one well designated to the pre-induced culture, one well devoted to the culture induced at time zero, and three wells filled with uninduced culture. The plate was returned to the 37°C incubator to shake at 230 RPM. Inducer was added to two wells of the uninduced cultures 20 and 40 minutes post-infection respectively, leaving one uninduced control. The experiment was ended with mechanical lysis of cultures and subsequent quantification of phage titers.

## Determination of proton motive force

*V. cholerae* containing the specified plasmids were grown to $OD_{600-tube}$=0.2. Inducers (125 µM IPTG and 187.5 µM theophylline) were added and cells were grown at 37°C with aeration for 30 minutes before 0.5 mL were pelleted at 5000 x *g* for 3 minutes and resuspended in 0.1 mL of phosphate buffered saline (pH 7.2; Gibco Life Technologies) containing 20 µM 3,3'-diethloxacarbocyanine iodide ($DiOC_2$(3); Sigma-Aldrich). Fluorescence measurements were completed in black 96-well half-volume plates (Corning) in the SpectraMax i3x (Molecular Devices) with 480(508) and 488(650) excitation(emission) wavelength settings.

## Plaque analysis

*V. cholerae* strains were grown to mid-log before plaquing assays. For EOP experiments, plasmid constructs were induced for 20 minutes prior to infecting and plating with antibiotic and inducer in the 0.5% LB top agar. For plaque edge analysis, plasmids were maintained with antibiotics but not induced prior to plating on 0.5% top agar containing antibiotics and inducer. Plates solidified at room temperature prior to incubation at 37°C. For spot plates, *V. cholerae* was mixed with 0.5% top agar prior to infection, vortexed, and poured onto an LB agar plate to solidify before 3 µL spots of phage dilutions were overlaid on the agar. Plates were allowed to dry prior to incubation at 37°C. To visualize plaques, plates were scanned on the EPSON Perfection V800 Dual Lens scanner.

## Western blots

Plasmid empty vector (EV) and FLAG-LidI[PLE 1] were grown to $OD_{600-tube}$=0.2 then induced with 1 mM IPTG and 1.5 mM for 40 minutes before 0.5 mL samples were taken. To observe expression during infection, strains were grown to $OD_{600-tube}$=0.3, infected with ICP1 MOI = 2 with 4 mL samples taken at the labeled timepoints. Samples were prepared and visualized as previously described (*McKitterick and Seed, 2018*). Briefly, samples were mixed with one volume of cold methanol, pelleted at 15,000 x *g* at 4°C for 15 minutes, washed with 1 mL cold PBS, and pelleted. Pellets were resuspended in PBS with XT sample buffer and reducing agent (Bio-Rad), vortexed, and boiled for 10 minutes before being run on 4–12% Bis-Tris SDS gels (Bio-Rad Criterion XT). Gels were transferred via the Trans-Blot Turbo (Bio-Rad) and visualized with rabbit-α-FLAG (1:3,000) primary and goat-α-rabbit-HRP conjugated secondary antibodies on the ChemiDoc MP Imaging System (Bio-Rad).

## CRISPR-Cas targeting of phage populations

*V. cholerae* harboring an empty vector control or LidI[PLE 1] plasmid were grown to $OD_{600-tube}$=0.15 before being induced and returned to the incubator to grow with aeration until $OD_{600-tube}$=0.3. At this point 150 µL of culture were added to 96-well plates and infected with ICP1 MOI = 0.1. After lysis 90 minutes post-infection, any remaining cells were mechanically lysed and the resulting phage population was plaqued on CRISPR-Cas (+) *V. cholerae* as previously described (*Box et al., 2016*). Briefly, Cas (+) *V. cholerae* harboring CRISPR array plasmids were induced 20 minutes before phage populations were titered on each specified *V. cholerae* strain. EOPs were determined by dividing the number of PFU/mL on *V. cholerae* containing spacers by the PFU/mL on the CRISPR-Cas (+) *V. cholerae* that did not contain a spacer against the phage (denoted as spacer 'none').

## Coinfection for homologous recombination

Coinfection experiments were completed in the same manner as the CRISPR-Cas targeting of phage populations described above with minor alterations: instead of infection with wild type ICP1, cultures

were coinfected in the plate with CRISPR*-Cas ICP1 and CRISPR-Cas* ICP1 each at an MOI = 0.01, observed for 200 minutes in the plate reader, and, after mechanical lysis, phage populations were plaqued on PLE (-) *V. cholerae* and PLE 1 *V. cholerae*. Control infections with only one phage never formed plaques on PLE 1 *V. cholerae.* The proportion of phages that successfully recombined was determined by the dividing the PFU/mL of each phage population on PLE 1 *V. cholerae* divided by the PFU/mL on PLE (-) *V. cholerae.*

## Bioinformatics

Transmembrane domains were predicted by conversion of all predicted open reading frames to amino acid sequence by CLC (*CLC, 2020*) before analysis with TMHMM Server v. 2.0 (*Sonnhammer et al., 1998*). PRALINE was used to create amino acid alignments of LidI (*Simossis and Heringa, 2005*). Homologs of TeaA (30% identity over 85% of the query) and ArrA (20% identity over 75% of the query) were identified with BLASTP (*NCBI NIH, 2019*) and arranged into phylogenetic trees as previously described (*McKitterick et al., 2019a*). Briefly, alignments were completed with MUSCLE v3.8.31 (*Madeira et al., 2019*) and a bootstrapped (n = 100) maximum-likelihood phylogenic tree was solved with PhyML 3.0 (*Guindon et al., 2010*). Default settings were used for amino acid sequences: automatic model selection with Akaike Information Criterion; SPR tree improvement with n = 10 random starting trees. Trees were visualized with FigTree (*Rambaut et al., 2019*).

## Acknowledgements

This work was supported by the National Institute of Allergy and Infectious Diseases [R01AI127652 to KDS]. KDS is a Chan Zuckerberg Biohub Investigator and holds an Investigators in the Pathogenesis of Infectious Disease Award from the Burroughs Wellcome Fund. Thanks to the Seed Lab for thoughtful discussions and help with special thanks to Zoe Netter, Caroline Boyd, Zach Barth, and Amelia McKitterick for reviewing drafts.

## Additional information

### Competing interests

Kimberley D Seed: is a scientific advisor for Nextbiotics, Inc. The other author declares that no competing interests exist.

### Funding

| Funder | Grant reference number | Author |
| --- | --- | --- |
| National Institute of Allergy and Infectious Diseases | R01AI127652 | Kimberley D Seed |
| Burroughs Wellcome Fund | Investigators in the Pathogenesis of Infectious Disease Award | Kimberley D Seed |
| Chan Zuckerberg Biohub | | Kimberley D Seed |

The funders had no role in study design, data collection and interpretation, or the decision to submit the work for publication.

### Author contributions

Stephanie G Hays, Conceptualization, Data curation, Formal analysis, Investigation, Methodology; Kimberley D Seed, Conceptualization, Funding acquisition, Project administration

### Author ORCIDs

Stephanie G Hays (iD) https://orcid.org/0000-0001-8829-2586
Kimberley D Seed (iD) https://orcid.org/0000-0002-0139-1600

Decision letter and Author response
Decision letter https://doi.org/10.7554/eLife.53200.sa1
Author response https://doi.org/10.7554/eLife.53200.sa2

## Additional files

### Supplementary files

• Source data 1. This spreadsheet contains the data used to create *Supplementary files 2* and *3*.

• Supplementary file 1. ICP1_2006_E gene product (gp) GenBank References. The gene products referred to in this work relate to open reading frames (ORFs) as noted in the 'Locus Tag Note'.

• Supplementary file 2. TeaA homologs. BLASTP was used to find homologs that share 30% identity with TeaA over 85% of the query. The GenBank ID, description, number of transmembrane domains (TMD) as predicted by TMHMM Server 2.0, and organism is listed for each homolog. Whether or not an ArrA homolog was found in the same organism is noted in the 'ArrA' column. Additionally, the adjacent upstream and downstream genes were analyzed for TMDs. GenBank descriptions are color coded. Due to the number of homologs analyzed, this table is only available as a spreadsheet as *Source data 1*.

• Supplementary file 3. ArrA homologs. BLASTP was used to find proteins with 20% identity to ArrA over 75% of the query. The GenBank ID, description, number of transmembrane domains (TMD) as predicted by TMHMM Server 2.0, and organism is listed for each homolog. Whether or not a TeaA homolog was found in the same organism is noted in the 'TeaA' column. Additionally, the adjacent upstream and downstream genes of each homolog were analyzed for TMDs. GenBank descriptions are color coded. The source data for this table is available in *Source data 1*.

• Supplementary file 4. Primer Table. Primers used in this work are provided with a description, identifier, and sequence.

• Transparent reporting form

### Data availability

All data generated or analysed during this study are included in the manuscript and supporting files.

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
