## [Decision Letter]

**Acceptance summary:**

In this work, Hays and Seed characterize a new phage lysis inhibition phenotype encoded by the ICP1 phage in the important human pathogen *V. cholerae*. Lysis inhibition, first discovered about 70 years ago, serves to increase phage burst size and protect phage progeny from adsorbing to cells that are already infected. Prior to this work, lysis inhibition had only been well-characterized in *E. coli* T-even phages, although it was also reported in coliphage N4. In these T4 phages, lysis inhibition is mediated by antiholins that inhibit the activity of phage holin proteins and slow progression towards cell lysis in the presence of high phage concentrations.

**Decision letter after peer review:**

Thank you for sending your article entitled "Dominant *Vibrio cholerae* phage exhibits lysis inhibition sensitive to disruption by a defensive phage satellite" for peer review at *eLife*. Your article has been evaluated by three peer reviewers, and the evaluation has been overseen by a Reviewing Editor and Wendy Garrett as the Senior Editor.

In this work Hays and Seed characterize a new phage lysis inhibition phenotype encoded by the ICP1 element in the important human pathogen *V. cholerae*. Lysis inhibition, first discovered about 70 years ago, serves to increase phage burst size and protect phage progeny from adsorbing to cells that are already infected. Prior to this work, lysis inhibition had only been well-characterized in *E. coli* T-even phages, although it was also reported in coliphage N4. In these T4 phages it is mediated by antiholins that inhibit the activity of phage holin proteins and slow progression towards cell lysis in the presence of high phage concentrations.

This work provides the first study of lysis inhibition outside of *E. coli*. Phage ICP1, a lytic phage that infects *V. cholerae*, is the predominant phage in cholera patient samples. The parasitic phage satellite PLE is integrated into the clinical *V. cholerae* isolates and it abolishes ICP1 phage production. When ICP1 infects, PLE is excised from the chromosome, replicates, and accelerates cell lysis; this results in release of PLE, and no infectious ICP1. One open question was how PLEs restrict ICP1, as no single PLE gene had been shown to be necessary for inhibition. Hays and Seed show that ICP1 inhibits lysis inhibition – they discover previously uncharacterized ICP1 genes with holin and antiholin activity (*teaA* and *arrA*), and they characterize a single PLE-encoded gene, lidI, which collapses ICP-mediated lysis inhibition. LidI is the first PLE-encoded ORF shown to negatively impact ICP1 phage yield. They show that all PLEs encode LidI, thereby revealing a conserved strategy to antagonize the phage lifecycle and inhibit progeny production.

There are a number of technical questions that need to be clarified, after which the reviewers feel confident that the authors have shown that a genuine LIN state is induced in ICP1 infections and aborted by PLE.

Major concerns

1) The bioinformatic analyses could be improved with a more systematic investigation of the TeaA and ArrA homologues. For example, the BLASTp hits for TeaA and ArrA are noted to be present "throughout marine phage and bacterial genomes". However, the figures appear to show that they are only found in Vibrio spp. The statement made in the paper makes it sound as though they are not restricted to Vibrio, and it should be made clear with respect to where these homologues are found. Also, an indication of how many homologues were found for each of the proteins and a table listing the hits (if this is a manageable number) with the associated ArrA homologues would be useful. It seems that not all *teaA* genes are associated with *arrA* genes. In cases where no ArrA homologue is identified, is there some other transmembrane-containing protein encoded next to *teaA* where *arrA* is expected to be?

– "the presence of ArrA homologues without TeaA homologues" – where is this data presented?

2) There are some confusing issues that should be resolved in the CRISPR-targeting experiments presented in the last section of the Results. In the absence of a spacer, there is approximately 10-fold difference in pfu/mL produced by LidI^PLE 1^ – and + samples. This is approximately the same difference that is noted between the LidI^PLE 1^ – and + samples in each of the three CRISPR targeting samples (Figure 6D). It's not clear to me how the conclusion that fewer phage progeny from LidI^PLE 1^
*V. cholerae* were able to overcome targeting than progeny from strains without LidI^PLE 1^ (subsection “LidI lowers phage yield”). It seems that the difference between phages produced in +/- LidI^PLE 1^ strains remains constant in the absence of CRISPR targeting or in the presence of the three spacers. Why is the targeting so poor for the spacers, particularly in the case of spacer A, which shows less than 10-fold decrease in phage plaquing? How were the three spacers chosen? Do the authors always find such poor targeting with CRISPR spacers and ICP1? I also had difficulty following the line of reasoning presented at the very end of the Results, centered on the reconstitution of the CRISPR system and the impact on evolution in infected cells.

3) There is some concern about the importance of both Figures 5 and 6. The authors have no real evidence that LidI is the LIN collapsing protein. It is an integral membrane protein with 2 TMDs and its overexpression will likely cause depolarization of the membrane and that would collapse LIN, just like DNP. Unfortunately, the lidI knockout has no phenotype. Regarding Figure 5, if one starts with low MOI, it takes longer to see LIN just because most cells are not infected. Recommend deleting these figures.

---

## [Author Response]

Major concerns1) The bioinformatic analyses could be improved with a more systematic investigation of the TeaA and ArrA homologues. For example, the BLASTp hits for TeaA and ArrA are noted to be present "throughout marine phage and bacterial genomes". However, the figures appear to show that they are only found in Vibrio spp. The statement made in the paper makes it sound as though they are not restricted to Vibrio, and it should be made clear with respect to where these homologues are found. Also, an indication of how many homologues were found for each of the proteins and a table listing the hits (if this is a manageable number) with the associated ArrA homologues would be useful. It seems that not all teaA genes are associated with arrA genes. In cases where no ArrA homologue is identified, is there some other transmembrane-containing protein encoded next to teaA where arrA is expected to be?"the presence of ArrA homologues without TeaA homologues" – where is this data presented?

The trees for ArrA and TeaA are in Figures 3E and Figure 3—figure supplement 1, respectively. Figure 3E shows the presence of ArrA homologs without TeaA homologs (squares without yellow dot). We have expanded the bioinformatics analyses on these homologs in the revised manuscript (available in Supplementary files 2 and 3). These newly added supplementary files include the GenBank Description and number of predicted transmembrane domains for each homolog as well as for the adjacent genes as per reviewer request.

We note that the homologs for TeaA are found throughout marine phage and bacterial genomes as shown in Figure 3—figure supplement 1. This figure shows that TeaA homologs are found in phages that infect Vibrionaceae (indicated by filled colored boxes) as well as in a number of bacteria (Pseudoalteromonas being the most common; signified by black box outlines) and multi-species metagenomic samples (signified in grey box outlines). We have clarified the statement to specify this is true for TeaA (and not for TeaA *and* ArrA as originally written). ArrA homologs are indeed limited to phages, specifically phages that infect the Vibrionaceae family (limited to members of the Vibrio and Entrovibrio genera) as we note in the text:

“ArrA yielded fewer homologs than TeaA (Supplementary file 3), however, using less stringent search parameters, we found that some organisms containing TeaA homologs also contain ArrA homologs though these were limited to vibriophages (Figure 3E and Supplementary file 2 and 3)”.

We do not expect all TeaA homologs to be associated with ArrA homologs because holins are fully functional without antiholins present. We think what is more interesting is that we find antiholins (i.e. ArrA homologs) in organisms without a holin (TeaA) homolog because antiholins function to regulate holins. As such, we displayed this information as yellow circles on both trees and clarified this in the text:

“This suggests that there are potential homologous lysis inhibition systems – complete with both holin and antiholin – present in phages other than ICP1. In contrast, the presence of ArrA homologs without TeaA homologs raises the question: what are antiholins doing on their own? Perhaps they have evolved functionality with holins divergent enough to no longer be considered homologous to TeaA under our search parameters, or they have been coopted for divergent functions much like holins (Mehner-Breitfeld et al., 2018; Saier and Reddy, 2015).”

2) There are some confusing issues that should be resolved in the CRISPR-targeting experiments presented in the last section of the Results. In the absence of a spacer, there is approximately 10-fold difference in pfu/mL produced by LidI^PLE 1^ – and + samples. This is approximately the same difference that is noted between the LidI^PLE 1^ – and + samples in each of the three CRISPR targeting samples (Figure 6D). It's not clear to me how the conclusion that fewer phage progeny from LidI^PLE 1^*V. cholerae* were able to overcome targeting than progeny from strains without LidI^PLE 1^ (subsection “LidI lowers phage yield”).

Reviewers are correct that LidI decreases the PFU/mL produced and this finding is the major result of figure panels 6A and B and Figure 6—figure supplement 1D. These panels demonstrate that LidI expression decreases the yield of phage from infected cultures. This is confirmed again in the samples with and without spacers targeting the phage in Figure 6D.

This effect is further quantified in Figure 6—figure supplement 4A where the efficiency of plaquing of LidI + strains in respect to LidI – strains is ~0.1 for all spacers and the control meaning that, though the absolute number of phages has been impacted, the frequency of escapes has not.

While the proportion of progeny phage that can overcome the spacer remains the same (Figure 6—figure supplement 4A), the total number of phage that can overcome a spacer from a LidI + infection is less than the total number of phage that can overcome a spacer from a LidI– infection (Figure 6D). For example, an average of ~6.13x10^7^ more phages per mL can plaque on spacer A from a LidI– infection in comparison to a LidI+ infection.

We have worked to make this clear in the text:

“For each spacer, fewer phage progeny from lidI^PLE 1^*V. cholerae* were able to overcome targeting than progeny from infections of strains without lidI^PLE 1^ (Figure 6D). This defect is due to the LidI-mediated decrease in the population of phages as the frequency of phage escaping stays the same (e.g. ~2 out of every thousand phages can overcome spacer C, Figure 6D). Consequently, because less phage are produced from lidI^PLE 1^-expressing *V. cholerae*, an order of magnitude fewer phages can overcome the spacer in the population (Figure 6—figure supplement 1).”

It seems that the difference between phages produced in +/- LidI^PLE 1^ strains remains constant in the absence of CRISPR targeting or in the presence of the three spacers.

This is exactly the case and is shown conclusively Figure 6—figure supplement 4A. Our interpretation of these data is in agreement with these comments and we have expanded the text as stated above that indeed the frequency of escapes remains the same +/-LidI.

Why is the targeting so poor for the spacers, particularly in the case of spacer A, which shows less than 10-fold decrease in phage plaquing? How were the three spacers chosen? Do the authors always find such poor targeting with CRISPR spacers and ICP1?

Spacer A shows ~10-fold drop in phage while spacers B and C show ~1000-fold drop. These decreases are within the range of previously published values and are linked to the frequency of escapes (Box et al., 2016). The additive effect that LidI has on the total number of phage that can overcome targeting is consistent across spacers. We chose these spacers from the lab’s collection and could add more if reviewers think it would be helpful hoewever we do not expect the results to change – spacers will work to varying degrees while LidI expression will decrease the population of phages that overcome the spacer by ~10-fold.

I also had difficulty following the line of reasoning presented at the very end of the Results, centered on the reconstitution of the CRISPR system and the impact on evolution in infected cells.

We have provided additional supplementary schematics to clarify the experimental set up and better contextualize these experiments for the reviewers. For Figures 6C and D and further explained in Figure 6—figure supplement 2, the CRISPR system discussed is encoded by *Vibrio cholerae* and targets the phage (“we exposed PLE (-) *V. cholerae* with and without lidI^PLE 1^ to ICP1 (MOI=0.1), collected the population of progeny phage, and looked for plaque formation on PLE (- *V. cholerae* encoding a Type I-E CRISPR-Cas system (Box et al., 2016); an expanded schematic of this experiment is available in Figure 6—figure supplement 2.”).

In contrast, for Figures 6E and F inactive CRISPR systems are encoded by two phage strains and target the PLE within the *V. cholerae* chromosome (“To test the impact of LidI^PLE^ 1 on this aspect of evolvability, we took advantage of the Type I-F CRISPR-Cas system found in ICP1 by engineering two ICP1 variants with nonfunctional CRISPR-Cas systems: one devoid of spacers against PLE with an inactive Cas1 preventing spacer acquisition (CRISPR*-Cas ICP1) (McKitterick et al., 2019) and the other lacking Cas2-3 (CRISPR-Cas* ICP1).”). These phage with inactivate CRISPR systems can only plaque on PLE+ strains if homologous recombination occurred reconstituting an active CRISPR system in the phage (“progeny phage were tested for their ability to plaque on PLE 1 *V. cholerae*, which is only possible if homologous recombination between the two variants restored a functional CRISPR-Cas system able to target PLE 1”).

These assays probe the evolution of the phage by measuring how many successful recombinants are formed in the presence and absence of LidI as shown in Figure 6—figure supplement 3. The key finding is that by limiting the population size as a whole, the number of phage that can evolve to overcome some selection pressure (either by mutations to escape CRISPR targeting or homologous recombination to restore a functional CRISPR-Cas system) is also limited.

3) There is some concern about the importance of both Figures 5 and 6. The authors have no real evidence that LidI is the LIN collapsing protein. It is an integral membrane protein with 2 TMDs and its overexpression will likely cause depolarization of the membrane and that would collapse LIN, just like DNP. Unfortunately, the lidI knockout has no phenotype. Regarding Figure 5, if one starts with low MOI, it takes longer to see LIN just because most cells are not infected. Recommend deleting these figures.

Figure 6 shows the decreased yield that results from LidI expression and the consequent collapse of lysis inhibition. This is an important finding in line with literature about T4 lysis inhibition and we observe the decreased phage production consistently across all tested conditions:

yield after phage infection in with LidI from both PLE 1 and PLE 4 induced at various times in respect to infection at high MOI (MOI=5) in Figure 6A and B

phage capable of overcoming *V. cholerae* encoded-CRISPR targeting after low MOI infection conditions (MOI=0.1) in Figure 6D

phage capable of recombining to reconstitute a functional phage encoded-CRISPR-system after low MOI infection conditions (MOI=0.01)) in Figure 6F

and we have now included three very low MOI infections of LidI expressing strains in Figure 6—figure supplement 1 (MOI=0.001, 0.0001, 0.00001) at the reviewer’s request

This finding was identified by the reviewers as important in the summary statement where, “LidI is the first PLE-encoded ORF shown to negatively impact ICP1 phage yield” and in the minor concern number 5 where a reviewer stated, “the authors designed a nice experiment to measure phage yield presented in Figure 6A where a specific high MOI was used. The two log decrease in yield is impressive under these conditions.” We believe this series of experiments marks the finding as consistent, reproducible, and significant, as such we do not agree to the recommendation to delete these figures.

In Figure 5, we do not claim that LidI is the protein that collapses lysis inhibition but instead that it is a protein sufficient to collapse lysis inhibition, which the data shows conclusively in Figures 4A, 5A, and Figure 6—figure supplement 1A-C. We further show that the PLE 4 LidI homolog is also sufficient to disrupt lysis inhibition in Figure 4E. We are careful to state that lysis inhibition collapse still happens in the absence of LidI (Figure 4A and E; subsection “PLE accelerates ICP1-mediated lysis”).

At this time, we can propose many molecular mechanisms for LidI activity: it likely could depolarize the membrane directly upon expression, which we show occurs after phage infection (Figure 4C) as the reviewer suggests, alternatively it could stabilize holin pore formation, destabilize the antiholin, inhibit holin-antiholin binding, or work through some other unforeseen mechanism. At this time we cannot distinguish between these hypotheses beyond stating that LidI expression alone does not depolarize membrane in the absence of phage infection (Figure 4C; subsection “PLE accelerates ICP1-mediated lysis”) yet still is sufficient to collapse lysis inhibition under the same induction conditions (Figure 4A and E).

Additionally, the reviewers communicated the following via email: The authors still leave the impression that LidI is the protein that collapses LIN. "LidI is the first PLE-encoded ORF shown to negatively impact ICP1 phage yield". This sounds like LidI actually does this but there is no evidence to this effect. Indeed, by earlier standards of phage genetics, we would conclude that it does not have this function, since there is no phenotype. The language still needs to be softened to avoid this false impression.

As stated above, we are careful not to say that LidI is the protein that collapses lysis inhibition in the context of PLE in the revised manuscript, but that it is sufficient to collapse lysis inhibition outside of the context of PLE. We conclusively show that LidI is the first PLE-encoded ORF demonstrated to negatively impact ICP1 phage yield (by orders of magnitude) in Figure 6 and Figure 6—figure supplement 1D. It is true that LidI could function differently in the context PLE, however, even if the function of LidI is a “fluke” in that LidI normally completes a different function in the cell, this “fluke” conveniently phenocopies the accelerated lysis we see in PLE+ cells. Although redundancy is perhaps not expected for mobile genetic elements with restricted genome size (like PLE), the most parsimonious explanation is that accelerated lysis is mediated by product(s) with redundant function, LidI being one of them (now stated in subsection “PLE accelerates ICP1-mediated lysis”). The earlier standards noted by the reviewer apparently do not allow for redundancy, nonetheless our investigation of this system is consistent with such redundancy and we believe this is even more of an interesting biological system because of it. The data demonstrating this redundancy has been added and can be found in Figure 4—figure supplement 3.

We have clarified language throughout the manuscript to impress upon the reader that the LidI characterization we have done is outside the context of PLE.

“Through lysis inhibition disruption a conserved PLE protein, LidI, can limit the phage produced from an infection, bottlenecking ICP1.”

“Subsequently, we discovered a single PLE-encoded gene we call lidI for lysis inhibition disruption that is sufficient to collapse ICP1- mediated lysis inhibition.”

“After confirming lidI^PLE 1^ expression during ICP1 infection, we next sought to characterize LidI^PLE 1^ function in PLE (-) *V. cholerae*. During infection with ICP1, LidI^PLE1^ was sufficient to recapitulate the PLE-mediated accelerated lysis phenotype.”

“Having demonstrated that LidI^PLE 1^ recapitulates PLE-mediated accelerated lysis, we wanted to determine if it was also necessary for this phenotype. Interestingly, however, when we deleted lidI^PLE 1^ from PLE 1 *V. cholerae* the lysis kinetics were unchanged (Figure 4A).”

“LidI^PLE 1^ can function through lysis inhibition disruption.”

“As LidI^PLE1^ is sufficient to accelerate lysis but does not phenocopy what we expect of a holin, we wanted to determine the mechanism of LidI^PLE 1^-mediated accelerated lysis in the absence of PLE.”

“Congruent with LidI^PLE 1^ disrupting lysis inhibition, lidI^PLE 1^ expression in PLE (-) *V. cholerae* does not change the efficiency of plaquing (EOP) by ICP1, an experiment that probes the number of successful initial infections at a low multiplicity of infection (Figure 5B). Consistent with this, the phenotypic change in plaque morphology expected of disrupted lysis inhibition is the loss of fuzzy plaque edges, which we see in PLE (-) *V. cholerae* expressing lidI^PLE 1^*in trans* (Figure 5C). It is important to note that these data showing that LidI disrupts lysis inhibition when expressed alone do not reveal the molecular mechanism underlying this activity or ensure that the gene serves the same function in the context of PLE, even though it successfully phenocopies PLE-induced accelerated lysis.”

“These data reveal LidI as the first PLE-encoded ORF that can singlehandedly negatively impact ICP1 phage yield.”

“Hence, we hypothesize that PLE-mediated accelerated lysis decreases the ability of ICP1 to evolve in the face of PLE. However, because our data support a model in which accelerated lysis is redundantly encoded, it is not currently possible to test the impact of delayed lysis on ICP1 evolution in the context of the PLE. We can, however, interrogate how the lidI-mediated collapse of lysis inhibition and concomitant decrease in phage production in PLE (-) *V. cholerae* constrains ICP1 evolution.”

“A single open reading frame, lidI, which is conserved through all five PLEs spanning the last 70 years, is sufficient to disrupt lysis inhibition and limit progeny phage populations when expressed outside its native context in PLE (-) *V. cholerae*.”

“These two opposing forces, ICP1 lysis inhibition and accelerated lysis by PLE through lysis inhibition disruption (which our data shows LidI is capable of doing in isolation and yet other undiscovered PLE-encoded mechanisms redundantly accomplish), act in the midst of the ongoing evolutionary arms race between *V. cholerae* and its parasites.”

Surprisingly, this is so important that the means to disrupt lysis inhibition and execute lysis for PLE’s own selfish benefit are apparently redundantly encoded within the PLE – LidI can collapse lysis inhibition in the absence of PLE, though DlidI PLE still shows accelerated lysis.

We also further state that PLE may encode redundant ways to accelerate lysis subsection “PLE accelerates ICP1-mediated lysis”: “This accelerated timescale could result from any combination of processes such as PLE deploying its own lysis machinery, PLE modulating the expression or stability of ICP1’s lysis machinery, or PLE inhibiting or collapsing lysis inhibition.” and show that inhibition of plaque formation and accelerated lysis phenotypes are robust to knocking out each individual ORF in Figure 4—figure supplement 3.

In respect to the importance of Figure 5, Figure 4 shows the reader that LidI is a conserved protein sufficient to cause accelerated cell lysis, that LidI is expressed during infection, and is not functioning as a holin in the absence of phage infection. The question that logically follows is how can LidI achieve acceleration of lysis? We do not specify molecular mechanism for the reasons explained above and instead check the obvious characteristics of lysis inhibition probing MOI (during low MOI infections the LIN state will be achieved later in time as described by the reviewer and in the final paragraph of subsection “PLE accelerates ICP1-mediated lysis”), checking the plaque edge phenotype, and efficiency of plaquing. The resulting data are consistent with disruption of lysis inhibition so much so that the reviewer raising this concern has already hypothesized the mechanism by which lysis inhibition is collapsed: depolarization of the membrane.